

**Spring distribution of shelled pteropods across the Mediterranean**
**Sea**
Roberta Johnson[1], Clara Manno[2], Patrizia Ziveri[1,3]
[1]Institut de Ciència i Tecnologia Ambientals, Universitat Autònoma de Barcelona, Barcelona,
Spain
[2]The British Antarctic Survey, Cambridge, United Kingdom
[3]Institució Catalana de Recerca i Estudis Avançats (ICREA), Barcelona, Spain
Correspondence to: Roberta Johnson (roberta.johnson@uab.cat)
**Abstract**
Shelled pteropods represent an excellent sentinel for indicating exposure to ocean acidification
(OA). Here, for the first time, we characterise spring pteropod distribution throughout the
Mediterranean Sea, a region that has been identified as a climate change hot-spot. The presence
of a west-east natural biogeochemical gradient makes this region a natural laboratory to
investigate how the variability in environmental parameters may affect pteropod distribution.
Results show that pteropod abundance is significantly higher in the eastern Mediterranean Sea
where there is a higher aragonite saturation state ($\Omega$ar), showing that distribution is positively
correlated with $\Omega$ar. We also observed a resilience of pteropods to higher temperatures and low
nutrient conditions, including phosphorous limitation. The higher abundance of pteropods in
ultra-oligotrophic conditions (eastern Mediterranean Sea) suggests that this organism can play
an important role as the prime calcifying zooplankton within specific oligotrophic regions.

**1. Introduction**

Thecosome pteropods are holoplanktic mollusks found in all major world oceans (Bednaršek
et al., 2012). This organism plays an important role in the biogeochemical cycle (Manno et al.,
2019) and in ocean food webs as bacterivores, (predominantly) herbivores, and as pray for
higher trophic levels (Conley et al., 2018). Pteropods are very susceptible to changes in
carbonate saturation state ($\Omega$) due to their aragonite shell, which is a comparatively highly
soluble form of calcium carbonate (Mucci et al., 1989), and they therefore represent an
excellent sentinel for indicating the impacts of ocean acidification (OA) (Bednaršek et al.,
2016; Manno et al., 2017).




In the Mediterranean, pteropod species diversity, abundance and distribution are currently
poorly documented. The studies that have incorporated information on pteropods in the
Mediterranean Sea focus mostly on whole zooplankton communities, with data collected via
different methods (i.e. net mesh sizes, collection depth) and in different regions (i.e. Ligurian,
Tyrrhenian, Adriatic, Balearic, Ionian, Cretan and Levantine Seas) (Andersen, 2002; Andersen
et al., 2004; Batistić et al., 2004; Fernández de Puelles et al., 2007; Mazzocchi et al., 1997).
There are only two studies within the Mediterranean that focus solely on pteropod ecology
(Howes et al., 2015; Manno et al., 2019) and both are limited to very restricted geographical
regions. To our knowledge, there is no study on pteropod abundance and distribution across
the Mediterranean Sea covering the whole basin and relatively large biogeochemical gradients.
This is limiting our knowledge of this important calcifying zooplankton.

The Mediterranean Sea has been identified as a climate change hot-spot (Giorgi, 2006) that is
particularly responsive and vulnerable to ecosystem changes (Lazzari et al., 2014). This region
is undergoing rapid changes as a result of climatic and non-climatic forcing (Cramer et al.,
2018) and is experiencing an increase in temperature that is exceeding global trends with a
current (as of 2018) annual mean temperature of 1.4°C above late-nineteenth-century level
(Cramer et al., 2018). Sea temperatures are expected to rise by 1.5-2°C by the end of this
century, at a rate faster than the global average (Lazzari et al., 2014). Sea surface pH is
predicted to decrease in line with the global average (approximately 0.3 to 0.4 units by 2100)
(Flecha et al., 2015b; Geri et al., 2014; Kapsenberg et al., 2017) or even exceed the global
average decrease (Hassoun et al., 2015). The impacts of climate change on marine systems will
be diverse and complex, with predicted disruptions to population dynamics, geographical
distributions and ecosystem functioning, as well as losses in biodiversity and species richness
(Bulling et al., 2010; Harley et al., 2006; Lacoue-Labarthe et al., 2016). It is essential to
improve our knowledge of key communities that are likely vulnerable to these changes in order
to determine how organisms and communities will respond to ocean conditions under climate
change.

The Mediterranean Sea has distinct biogeochemical regions that cross natural environmental
gradients, with the shallow Strait of Sicily splitting the Mediterranean into east and west
(Rohling et al., 2009). The eastern Mediterranean is characterised by higher temperatures and
salinities than the western basin, which consists of Atlantic water entering from the Gibraltar



Strait that are modified moving eastward (Rohling et al., 2009). Surface-water circulation is
mainly driven by thermohaline forcing as well as wind stress (Robinson and Golnaraghi, 1994).
Using data collected from the MedSeA cruise (2013), the average Ωar (saturation state of
aragonite) in the top 200m of the water column gradually increases from approximately 2.7 in
the Atlantic to approximately 3.6 in the Eastern Mediterranean, meaning these waters are
saturated with respect to aragonite. Phosphate and nitrate have higher concentrations in the
west of the Mediterranean, with a sharp decrease moving to the east of Mediterranean, which
is typified by phosphorus limitation and low concentrations of nitrate (oligotrophic system)
(Krom et al., 1991). These stark changes in marine environmental parameters from west to east
make the Mediterranean Sea a natural laboratory to investigate how the variability in
environmental factor is affecting species distributions.

This study aims to investigate pteropod distribution across large spatial scales, such as the
Mediterranean Sea, which has been identified as a gap within pteropod research, particularly
in relation to understanding how these populations will respond to climate change (Bednaršek
et al., 2016). In this region, the presence of a West-East natural environmental gradient enable
us to investigate the interaction between pteropod distribution and environmental parameters
during the spring season. We also present the relationship between pteropods and another major
group of planktic marine calcifier, foraminifera (single-celled, calcareous zooplankton).
Foraminifera were collected during the same research cruise campaign (Mallo et al., 2017) and
are therefore directly comparable with this study on thecosome pteropods.

**2.  Materials and methods**

Samples were collected from the Mediterranean Sea during the MedSeA (Mediterranean Sea
Acidification in a Changing Climate) cruise from May 2$^{nd}$ to June 3$^{rd}$, 2013 (Fig. 1) (Ziveri and
Grelaud, 2015). The research cruise covered the majority of Mediterranean sub-basins, starting
from the Atlantic Ocean crossing the western basin and moving through the Levantine Basin
in the east, and then again starting from the north-eastern Ionian Sea, moving to the northern
Algero–Provençal basin in the east (Fig. 1).

**2.1 Hydrological and chemical collection analyses**



Temperature, salinity, oxygen and fluorescence were obtained from the correspondent
conductivity-temperature-depth (CTD) stations using an ITS-90 and an oxygen sensor SBE 43
and considering the upper 200m towing depth. The overall accuracy for temperature is ±
0.001°C and ± 0.0003 for salinity. To determine the seawater carbonate system, samples for
total alkalinity (AT) and dissolved inorganic carbon (DIC) were collected from the top 200m
of the water column (~5m, 10m, 20m, 40m, 50m and every 25m thereafter until 200m) (see
Goyet et al. (2015). Methods for the analysis of water chemistry (total alkalinity and dissolved
inorganic carbon) have been described in Gemayel et al. (2015; Hassoun et al., 2015). Ocean
chemistry data were input into the program CO2sys to calculate pH, $p$CO$_2$, aragonite saturation
($\Omega$ar) and [CO$_3^{-2}$] using the equilibrium constants of Mehrbach et al. (1973) refitted by Dickson
and Millero (1987) . Photosynthetically active radiation (PAR) was measured at the beginning
of each tow. Surface chlorophyll $a$ concentration was obtained from MODIS Aqua L2 satellite
data (NASA Goddard Space Flight Centre, 2013; Fig. 1). The nutrient concentrations
(phosphate [PO$_4$] and nitrate [NO$_3$]) were obtained using OGS (Italian National Institute of
Oceanography and Experimental Geophysics) and analysed with a Bran+Luebbe3
AutoAnalyzer (see Grasshoff et al. 1999) and D'Amario et al. (2017) for a detailed
methodology of the nutrient analysis).

**2.2 Pteropod sample collection and analyses**

123        Sampling for pteropods was conducted using BONGO nets (mesh size 150µm, 40cm

diameter) from the surface to approximately 200m depth. A flow meter attached to the ring of
the net was used to determine the volume filtered through the net. Please refer to Supplementary
Table 1 for information pertaining to the date, time, location, environmental parameters and
volume of water filtered in the plankton tow for each sampling station. From these tow samples,
pteropod abundances were determined for each station. Samples were stored in 500ml
polycarbonate bottles and kept at 4°C in the dark. pH was measured in all the samples, at the
beginning, during and the end of the storing period to ensure that the state of the pteropod shells
were not affected by the preservation technique. Pteropod abundance was determined for each
station and species were identified and counted using a Leica z16 APO binocular light
microscope. Pteropod abundance within the water column was calculated as individuals per
cubic meter (ind. m$^{-3}$). Pteropods were grouped into four target families: Heliconoididae,
Limacinidae, Cavoliniidae and Cresedai; and further into seven target species: *Heliconoides*
*inflata, Limacina trochiformis, Limacina bulimoides*, *Cavolina inflexa, Creseis acicula,*



*Creseis concia* and *Styliola subula.* The online plankton portal (www.planktonportal.org) was
used to aid in the identification of pteropods to species level.

**2.3 Statistical methods**

142        All environmental parameters used in the analyses were averaged from 5-200m depth.

Using the environmental parameters (temperature, salinity, oxygen, fluorescence, $NO_3$, $PO_4$,
pH, $pCO_2$, $CO_3^{-2}$ and $\Omega$ar) a principle component analysis was conducted (PCA, varimax
rotation). After an initial analysis, PAR was removed as it did not significantly contribute to
the variation of environmental parameters. Based on the PCA results, a Kruskal-Wallis Test
was used to determine any significant differences in total and individual species abundance
between western stations (1-7a and 19-22) and eastern stations (9-'16-18'). A parsimonious
canonical correspondence analysis (CCA) was used to determine the significant environmental
parameters affecting pteropod species composition. Using Factor 1 and Factor 2 values from
the PCA and the environmental parameters for each station, alpha values were obtained to
conduct a binary logistic regression model (BLRM) which predicts the odds of having a low-
density station (classified as stations with >1 ind. $m^{-3}$ pteropod). Stations were binned
according to the total density (13 high density stations, 7 low density stations) and were
modelled against Factor 1 and 2, as well as each environmental variable. Pearson's correlation
coefficients were calculated to determine if there were any significant relationships between
the environmental parameters and total and individual species abundance. The CCA was
analysed using R version 3.6.0 and all other statistical analyses were performed using IBM
SPSS v23.

**3.  Results**

The mean absolute abundance of pteropods collected in the Mediterranean Basin was 1.27 ±
1.62(SD) ind. $m^{-3}$ (STable 2). The highest abundance was recorded in the Otranto Straight
toward the southern end of the Adriatic Sea with 5.21 ind. per $m^{-3}$ (STable 2, Fig. 2). There
were no pteropods sampled in the Northern Alguero-Balear region (station 20), and the lowest
mean standing stock was recorded at the Catalan-Balearic Station, off the coast of Spain
(STable 2, Fig. 2). The eastern Mediterranean Sea had higher overall pteropod abundances with
an average abundance of 2.13 ind. $m^{-3}$, which was approximately 5x greater than the average



pteropod abundance in the western basin (0.42 ind. m$^{-3}$), where the lowest abundances were
recorded.

A total of five different pteropod families and 11 species were identified. Limacinidae was the
most abundant family (77.7%), which dominate the eastern part of the Mediterranean basin,
followed by Creseidae (15.4%) and Cavolinidae (6.8%). *Heliconoides inflata* was the most
abundant species in the Mediterranean (29.4%) and recorded at all stations, followed by *L.*
*bulimoides* (23.8%) and *L. trochiformis* (23.2%). The three species in Limacinidae followed a
similar distribution pattern with high abundances in the Ionian Sea and the Antikythera Strait
and lower abundances to the south of the Balearic Sea. *Cavolina inflexa* (6.8%), *C. acicula*
(7.7%) and *Creseis conica* (6.8%) all presented the highest abundance in the Eastern Ionian
Sea while *S. subula* (0.68%) abundance was at a maximum in the east of Levantine basin
(STable 2). Specimens within the target families that were unidentifiable to species level made
up 1.4% of the total abundance.

**3.1 Relationship with environmental parameters**

The Kruskal-Wallis Test confirmed significantly greater total pteropod abundance and
abundance of *H. inflata, L. trochiformis, L. bulimoides* and *C. inflexa* in eastern stations
(STable 3). The results of the Pearson's Correlation show that total pteropod abundance over
the whole Mediterranean Sea was positively correlated with $\Omega$ar, $O_2$, pH, salinity and
temperature and negatively correlated with $NO_4$, $PO_3$ and $p$CO$_2$ (STable 3).

The PCA performed on the environmental parameters reveals separation between stations in
the east and the west of the Mediterranean Basin (Fig. 3). The western basin is characterised
by greater nutrient content, lower temperatures and salinities (Fig. 3). The eastern basin is
characterised by ultra-oligotrophic conditions as well as higher temperatures, salinities and
aragonite saturation (STable 1). Factor 1 and 2 together explain approximately 77.55% of the
variation in the environmental parameters. Factor 1 explains approximately 58.08% of
variation in environmental parameters. This factor exhibited positive loadings for temperature
and $O_2$ (and to a lesser extent $\Omega$ar and salinity) and a negative loading for the nutrients $PO_4$ and
$NO_3$ (and to a lesser extent fluorescence). Factor 2 explains approximately 19.47% of variation
in environmental parameters. This factor is characterised by positive loadings for pH, salinity
and $\Omega$ar, and a negative loading for $p$CO$_2$. Factor 2 is interpreted as the carbonate system in



the Mediterranean Sea, with a higher pH and $\Omega$ar in the east compared to the west. Abundance scores for total pteropod and individual species counts were overlaid onto the PCA plot (Fig. 3 and SFig. 1). Stations in the eastern Mediterranean generally have a greater total abundance of pteropods and are more positively associated with Factor 1 and 2; namely, greater temperatures, salinities, pH levels, $\Omega$ar and $O_2$ (Fig. 3). In general, the western stations have a lower total abundance and are more positively associated with $p$CO$_2$, nutrients and greater nutrient availability.

The parsimonious CCA reveals that temperature, pH, $\Omega$ar and salinity affect pteropod community composition in the Mediterranean and that species group by family, with species belonging to Limacinidae showing a positive relationship with pH, $\Omega$ar and salinity, and *H. inflata* showing a strong positive correlation with temperature (Fig. 3). Species in Creseidae also group together yet do not correlate positively with temperature, pH, $\Omega$ar and salinity (Fig. 3), reflecting the lower proportion of total abundance for this family in the east of the Mediterranean. *Cavolina inflexa* does not group with the other families, and the results do not indicate a positive correlation with any environmental factor.

The BLRM predicts (90% CI) that an increase in temperature and, to a much greater extent $\Omega$ar, will increase the odds of having a station with a total abundance >1 ind. m$^{-3}$. Increasing nitrate will reduce the odds of having a station with a total abundance >1 ind./m$^{-3}$ (STable 3). These results are likely influenced by the higher abundance of family Limacinidae in the east of the Mediterranean Sea. The BLRM correlates well with the CCA results in that pteropod community composition is driven by a similar suite of environmental factors, namely $\Omega$ar, pH, salinity, and to a lesser extent, temperature.

## 4. Discussion

This study shows that springtime shelled pteropod distribution in the Mediterranean Sea is positively correlated with $\Omega$ar saturation across the whole basin. We also found that the variability in $\Omega$ar influences community composition and that changes to the level of $\Omega$ar is likely to cause changes to pteropod populations. These results are driven by the positive relationship of $\Omega$ar with the most abundant family Limacinidae (76.4% of total abundance) and suggest that overall, pteropod distribution is strongly dependent on $\Omega$ar distribution.



The energetic costs associated with calcification indicate that calcifying organisms will spend
more energy on building their shells under low $\Omega$ar conditions (Waldbusser et al., 2015). A
recent study found that under reduced $\Omega$ar (control - 2.8; reduced - 2.1 $\Omega$ar), calcification in *H.*
*inflatus* decreased and metabolic rate increased (Moya et al., 2016), revealing that even in
oversaturated conditions, variability in $\Omega$ar can generate stress. Small variations in over
saturated aragonite conditions may incur an energetic cost (however small), which can result
in variations in pteropod geographical distribution. Species distribution often reflects
environments containing their optimal parameters in order to optimise energy expenditure
(Parsons, 1990) and our results show that pteropods have a preference for areas with higher
$\Omega$ar in which they will likely devote less energy to building their shell and be under less
metabolic stress. A study in the Mediterranean at $CO_2$ vents in the Gulf of Naples investigated
pteropod calcification and abundance along an $\Omega$ar gradient (1.9-2.7) (Manno et al., 2019).
Similar to our study, there was a positive correlation of *Creseis conica* abundance with $\Omega$ar in
oversaturated conditions.  The focus of this paper however, was to a very small geographic
region in the Tyrrhenian Sea and involved a non-static system ($CO_2$ vent), whereas in the
present study, we utilised stronger gradients of multiple environmental parameters over a large
geographic scale. The findings in our study indicate that springtime pteropod distribution in
the Mediterranean is strongly dependent on $\Omega$ar distribution.

The $\Omega$ar in this study ranged from 2.68 in the west of the Mediterranean to 3.61 in the east (Fig.
4A). It is difficult to determine seasonal or inter-annual variability for $\Omega$ar within the
Mediterranean, as studies on carbonate chemistry are either at a single time point over
latitudinal gradients (Álvarez et al., 2014; Millero et al., 1979; Schneider et al., 2007) or a mid
to long-term data series at a single coastal location (Ingrosso et al., 2016; Kapsenberg et al.,
2017). In the case of the long-term data series by Ingrosso et al. (Ingrosso et al., 2016) in the
Gulf of Trieste in the north Adriatic Sea and Kapsenberg et al. (Kapsenberg et al., 2017) in the
Ligurian Sea, samples for $\Omega$ar analysis were collected from shallow coastal locations (~15m
and ~75m respectively) and as such are not reflective of the open sea environments where
sampling in our study was undertaken. Seasonal variation of $\Omega$ar in open ocean waters such as
the North Atlantic (upper 50m and 100m depths; $\Omega$ar 0.4-0.6) (Kim et al., 2015) is likely more
reflective of the seasonal fluctuations experienced in deeper Mediterranean waters, and as such,
the seasonal variability in $\Omega$ar that pteropods are experiencing is not in the range of the
geographical variability within the Mediterranean Sea (up 200m). As $\Omega$ar is the biggest driver
of pteropod distribution, we suggest that the general population trend in this study will be





maintained throughout the year, as the eastern Mediterranean will be consistently higher in $\Omega$ar
and therefore more energetically favourable than the west in terms of calcification.

We found that to a lesser extent, temperature and $NO_3$ are also drivers of spring pteropod
distribution within the Mediterranean sea, with increases in temperature and decreases in $NO_3$
predicted by the BLRM to slightly increase the odds of higher pteropod abundance. In the
Mediterranean, sea surface temperatures vary by about $10^{\circ}$C over the year, with the north-
western Mediterranean have a winter average around $10^{\circ}$C and a summer average of $21^{\circ}$C,
while the south-eastern Mediterranean winter average is around $15^{\circ}$C and the summer around
$26^{\circ}$C (Rohling et al., 2009) with a consistent west to east gradient (Fig. 4C). Most of the
recorded temperatures in this study will be experienced at some point over the course of the
year for all pteropod populations, except for the eastern Mediterranean during summer, which
is warmer than the west year-round (Rohling et al., 2009).  In light of the ongoing warming of
the Mediterranean Sea, further studies should address the potential implication connected to
the observed positive relationship between pteropod abundance and temperature.

For $NO_3$, there is a well-known east-west gradient of oligotrophy in the Mediterranean (Fig.
4B) that does not have much annual variability(Pasqueron de Fommervault et al., 2015). In our
study, pteropod abundance was ~5x greater in the ultra-oligotrophic eastern Mediterranean.
The negative correlation of pteropod abundance with favourable nutrient conditions (in
particular nitrate) is unexpected, as pteropod global distribution has been previously correlated
with high productivity and nutrient content (Bednaršek et al., 2012; Burridge et al., 2017). A
study of pteropod distribution along a longitudinal gradient in the Atlantic Ocean (Burridge et
al., 2017) found that areas with the highest biomass corresponded to areas with the highest
concentrations of chlorophyll *a* (Huot et al., 2007). However, because in Burridge's study
chlorophyll *a* was correlated with temperature, it is unclear which of the two variables was the
main driver of pteropod distribution. As only two environmental parameters were investigated,
Burridge's study is limited in its' ability to determine how other biologically relevant
parameters for pteropods affect their distribution, such as $\Omega$ar. In the Atlantic Ocean over a
range spanning more than the 45ºN to 45ºS in Burridge's study, the upper 200m is
supersaturated with respect to aragonite (Jiang et al., 2015) and there is little variability in $\Omega$ar
(Takahashi et al., 2014). If $\Omega$ar is a primary driver of pteropod distribution, as indicated by this
study, when waters are supersaturated and there is little variability in $\Omega$ar, other factors such
as temperature or nutrients may be secondary driving forces affecting pteropod abundance, as

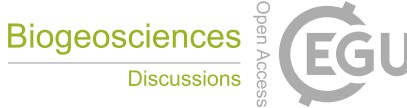

seen in Burridge's study. As we found that $\Omega$ar is the main driver of pteropod distribution in
the Mediterranean, it is likely that the negative correlation with $NO_3$ is not directly causing a
negative effect on pteropod abundance and distribution. Pteropod abundances are higher in the
oligotrophic and highly aragonite-saturated eastern basin, resulting in a negative correlation
with $NO_3$.

We are aware that this study is only a snapshot in time, which limits our ability to predict the
distribution of pteropods within the Mediterranean over large temporal scales. However, this
study represents the first description of pteropod distribution along the whole Mediterranean
basin and their relation with relevant environmental parameters. On the basis of the observed
correlation between pteropods and the natural geographical variability of $\Omega$ar, we suggest that
future research should focus on long-term, Mediterranean-wide monitoring to detect potential
changes of pteropod populations.

**4.1 Pteropod and foraminifera interaction**

Planktic foraminiferal abundance and distribution presented in Mallo et al. (2017) were
compared to the pteropod data from this study. The tow samples from Mallo et al. (2017) were
collected during the same cruise and within the same nets as the pteropods of the present study,
allowing a direct comparison of the two groups of key planktic calcifiers.

Spring pteropod and foraminiferal abundances present an opposite pattern distribution between
the west and east of the Mediterranean (Fig. 5). A clear difference in the composition of
pteropod and foraminiferal communities is evident in the CCA (Fig. 4), showing foraminiferal
communities are positively related to fluorescence and $NO_3$, and pteropod communities
positively related to temperature, pH, salinity, $O_2$ and $\Omega$ar. Pteropod abundance is distinctly
greater in the eastern ($\overline{x}$=2.13 ind. $m^{-3}$) Mediterranean than in the west ($\overline{x}$=0.47 ind. $m^{-3}$), while
foraminifera populations are characterised by higher abundances in the west ($\overline{x}$=1.87 ind. 10
$m^{-3}$) than in the east ($\overline{x}$=0.96 ind. 10 $m^{-3}$) (Fig. 5). There was a significant correlation in
abundance between western Mediterranean stations (r=.757*), but not the eastern stations.
Although total abundance is distinctly greater for pteropod populations across the entire
Mediterranean, abundance for both organisms follows a similar pattern in the western stations
(Fig. 5).





Experimental evidence indicates that planktic foraminifera are vulnerable to OA conditions
under which they experience reduced calcification and an increased metabolic rate (Davis et
al., 2017; Lombard et al., 2010; Manno et al., 2012). Foraminiferal distribution and abundance
appear to be driven more by nutrient levels than carbonate saturation levels, as suggested in
Mallo et al. (2017). The skeleton of foraminifera is made of calcite, a less soluble form of
calcium carbonate than aragonite (Mucci et al., 1989; Subhas et al., 2018). Both pteropods and
foraminifera are sensitive to OA due to their carbonate skeletons, however it is likely that
foraminifera are less sensitive to carbonate variability due to the lower solubility of their calcite
skeleton compared with aragonite. Foraminifera have been shown to be nutrient limited (Gregg
and Casey, 2007; Schiebel et al., 2004), and Mallo et al. (2017) suggested that the lower
foraminiferal abundance in the eastern Mediterranean (ultra-oligotrophic sector) results from
reduced reproduction due to limiting food resources.

Conversely, nutrients and food availability do not appear to be a limiting factor for pteropods
and their abundance is ~5x higher in the oligotrophic sector of the Mediterranean than the west.
The majority of pteropods belong to the Thecosome order and have a unique feeding method
involving the production of a large mucous web that is suspended in the water column, which
passively entraps organic particles and motile organisms, enabling them to filter water at high
rates (Conley et al., 2018). To ingest the material collected by the mucous webs, they draw the
webs into their mouth via ciliary action, a feeding method that may allow them to overcome
low food condition due to their ability to capture and filter through relatively large amounts of
organic matter (Hamner et al., 1975). In contrast to foraminifera, pteropods are also able to
actively swim (Hamner et al., 1975; Lalli and Gilmer, 1989) and potentially search for more
favourable food conditions. Further, the survival of pteropod *H. helicina* (Limacinadae family)
does not appear to be affected when expose to prolonged starvation (7 days) (Busch et al.,
2014). We suggest that, in the western Mediterranean, pteropods are more able to adapt to low
food availability than foraminifera which may be due to a combination of both their feeding
method and their ability to withstand starvation.

Thus the difference in the pattern of distribution between foraminifera and pteropods in the
eastern Mediterranean may be due to the unfavourable oligotrophic conditions for foraminifera
and that pteropods are more dependent upon higher $\Omega$ar than nutrient levels. In the west, with
relatively higher concentrations of nutrients, $\Omega$ar is lower, accounting for the reduced
abundance pteropods, with foraminiferal and pteropod abundance following the same pattern,



as a secondary factor of importance for pteropods, namely nutrients, affects their distribution
here. On the other hand, in the eastern oligotrophic conditions, pteropod abundance is distinctly
greater than foraminifera, as it is more strongly driven by the higher $\Omega$ar saturation, and
foraminiferal and pteropod abundance no longer follow the same pattern, as the region is very
nutrient poor, negatively affecting foraminiferal abundance.
There have been only a handful of studies that investigate the relationship between pteropod
and foraminiferal communities. A multi-decadal study of calcareous holo-zooplankton at two
sites off the coast of Southern California and Central California (1951-2008) showed no
relationship between foraminifera or pteropod abundances (Ohman et al., 2009), unlike our
study. In the Gulf Stream, Sargasso Sea and the Gulf of Mexico, pteropod and foraminifera
densities followed a similar pattern, with density decreasing closer to oligotrophic conditions
(Casey et al., 1979). In Schiebel et al. (2001), foraminifera and pteropod production in the
North Atlantic was positively correlated with chlorophyll-*a*. The correlation of abundance for
pteropods and foraminifera with nutrients in Casey et al. (1979) and Schiebel et al. (2001) are
similar to the findings for the west of the Mediterranean in our study. These studies did not
incorporate the carbonate system and our study is the first to utilise a complete spectrum of
environmental parameters to investigate the relationship between pteropods and foraminifera.

**5.  Conclusions**

The relationship of shelled Mediterranean pteropods to the natural variability in $\Omega$ar adds to
the growing body of support for using pteropods as a bioindicator of changes in ocean
carbonate chemistry (Bednaršek et al., 2019; Manno et al., 2017). Overall, we suggest that the
higher abundance of pteropods in the eastern than the western Mediterranean sea is a
combination of higher $\Omega$ar that is energetically favourable for calcification and their unique
method of feeding and/or ability to withstand starvation, allowing them to fill the ecological
role in the east of the Mediterranean as the prime calcifying zooplankton. With the increase in
OA, the positive correlation between $\Omega$ar and pteropod distribution stresses the need for
regular, long-term monitoring of pteropod populations throughout the entire Mediterranean
Sea. This study provides new insight into the ecology of shelled pteropods and foraminifera
and highlights that, in specific oligotrophic regions, ecological competition could also play an
important role in regulating the pelagic calcifier zooplankton communities.



**Acknowledgements**

We thank the captain and crew of the Spanish R/V Ángeles Alvariño for supporting the
sampling of this study, M. Acevedo and U. Tilves for sample collection, E. Xicoy Espaulella
for helping with sample processing. This work was funded by the Spanish Ministry of
Economy and Competitiveness (MINECO) (CALMED project - CTM2016-79547-R). Roberta
Johnson received a fellowship from MINECO (FPI/BES-2017-080469). This work is
contributing to the ICTA "Unit of Excellence" (MINECO, MDM2015-0552) and the MERS
research group of the Generalitat de Catalunya (2017 SGR-1588).

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

**Figure Legends**

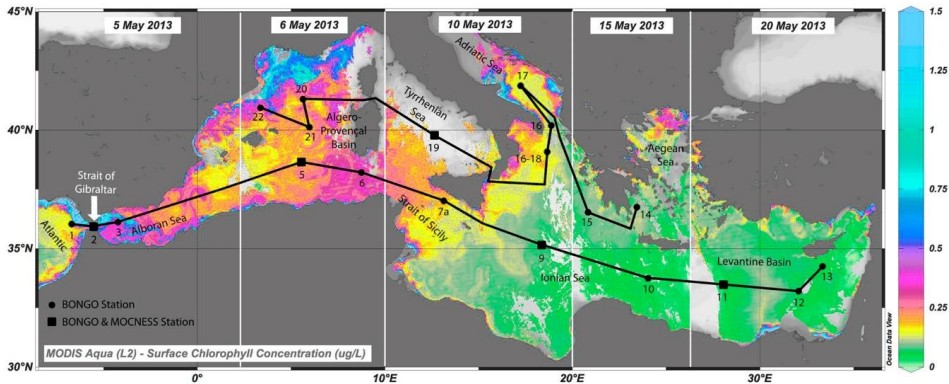





**Figure 1.** The sampled stations with BONGO nets are indicated by dots. The numbers in the
picture represent the station codes: First leg: 1 to 13, second leg: 14 to 22. The colour scale
represents the values of surface chlorophyll concentration (in µg/l), retrieved from *MODIS*
*Aqua (L2)*, from the closest day to the start of the first transect. Figure made using Ocean Data
View. Also in Mallo et al. (2017).



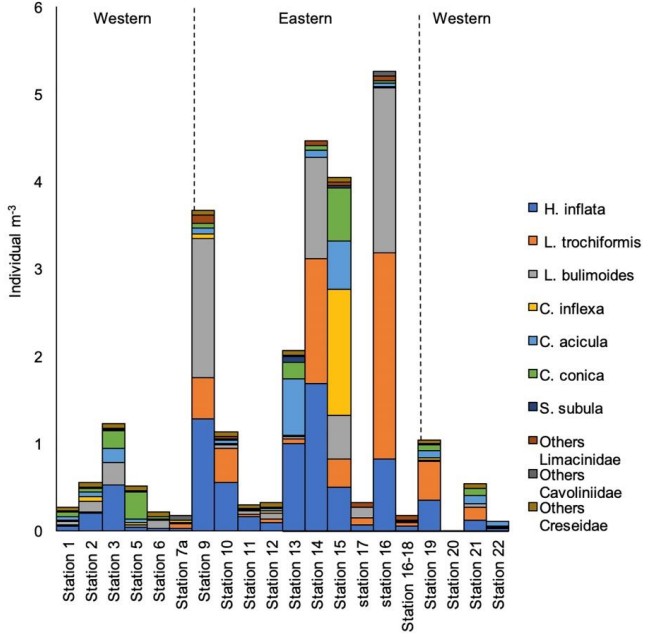



**Figure 2.** Absolute abundance of planktic pteropods from stations 1-22 on the MedSeA cruise.
The category of 'Others' for each family includes specimens that were not a target species in
this study or that were unidentifiable to the species level.



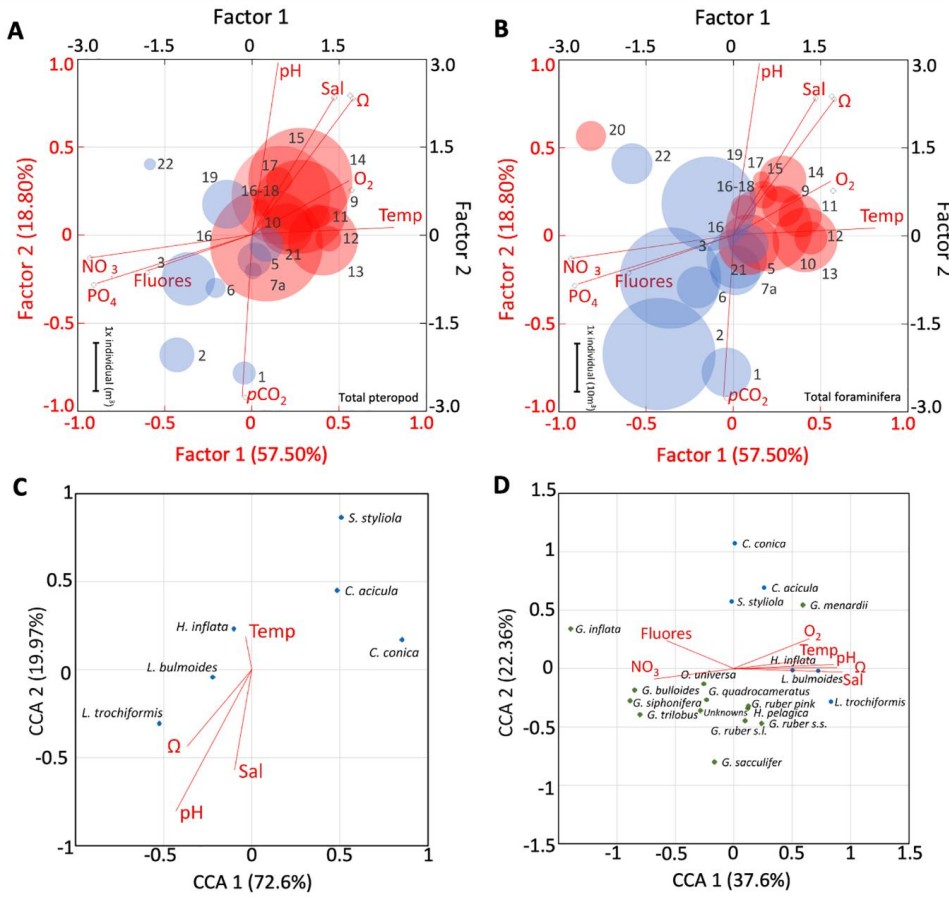

**Figure 3.** PCA graphs of environmental factors overlaid with absolute abundance values on station scores of **A** pteropods at all stations (ind. per m$^{-3}$) and **B** foraminifera at all stations (ind. per 10m$^{-3}$). The red axes are associated with PCA coordinates and the black axes are associated with the station coordinates. CCA graphs indicating **C** pteropod community composition and **D** pteropod (blue dots) and foraminifera (green dots) community composition. Family Cavoliniidae (*Cavolinia inflexa*) is not indicated on the graph as they were not related to any environmental factor or species groupings. Coordinates for *C. inflexa* in **C** (1.55, 0.59) and **D** (0.63, 2.33).


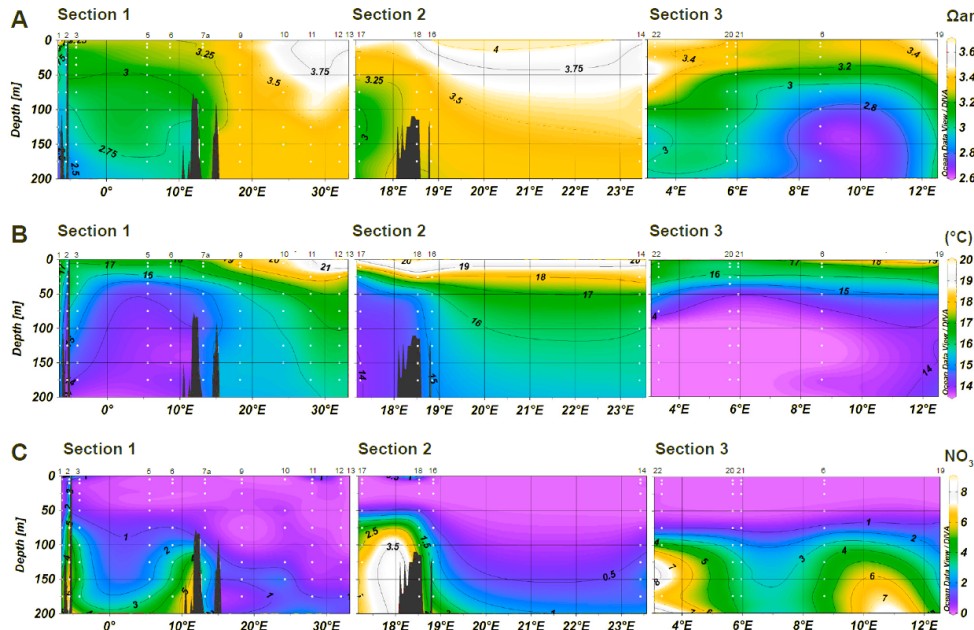

**Figure 4.** Depth profiles from 0-200 m depth highlighting the gradients of **A** Aragonite saturation (Ωar) **B** Nitrate (NO₃) and **C** Temperature (Cº) from the west to the east of the Mediterranean Basin (Section 1), within the Adriatic Sea (Section 2) and in the North-Eastern Mediterranean (Section 3) for factors affecting abundance and the community composition of pteropods. Figures made using Ocean Data View.

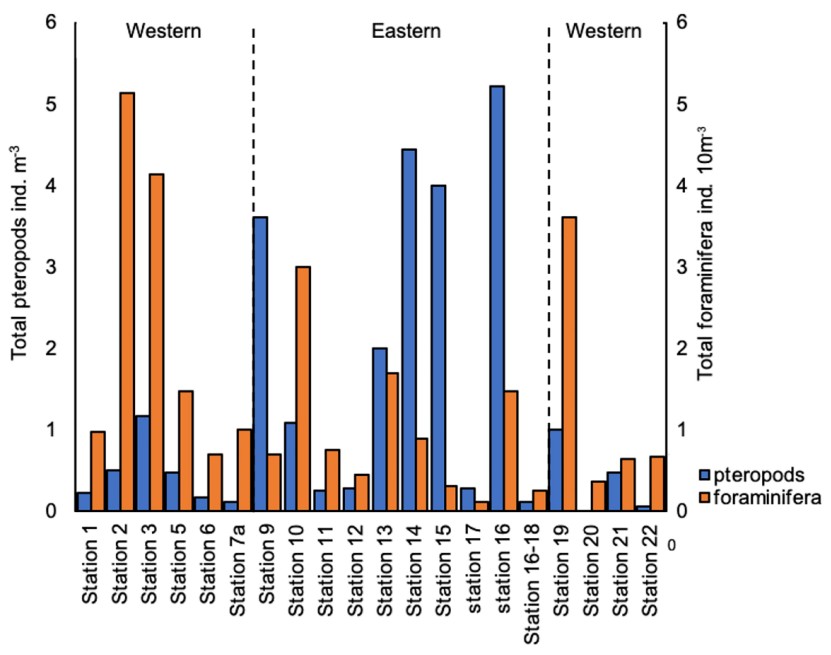

669

670

**Figure 5.** A comparison of absolute total abundance for pteropods and foraminifera at each station. Note that the scale of foraminifera abundance is distinctly lower than pteropod abundance (ind. 10 m$^{-3}$ and ind. m$^{-3}$ respectively), however this graph serves to illustrate similarities and differences in patterns of abundance.