# Peer review of "Spring distribution of shelled pteropods across the Mediterranean"

_Biogeosciences, 2020_

## Referee Comment (RC1) · Anonymous Referee #1 · 16 Mar 2020

Whilst this study provides an interesting dataset, there are significant issues in the design of the study and the analysis of the results that bring into question the findings.

Whilst the dataset used in this paper is definitely a useful resource, I regret that I do not think that it can be used in the way that the authors have outlined. My impression is that the sampling was not designed with the study question in mind, rather that the dataset already existed, and the study undertaken as an attempt to use it in some way. As a result, there are several issues with the use of the data to infer environmental controls on pteropods. I understand the limitation of field studies and appreciate the time, effort and resource required to acquire samples however, I do not feel that the dataset can be applied in this way to answer this study question.

The sample depth of 200 m is relatively shallow for sampling many of the species. The

early work of Rampal (1975) very clearly showed that many species exhibit seasonal depth preferences, with some populations spending many months of the year at depth in excess of 700 m. I do appreciate the difficulties in accessing this work as it is an unpublished thesis (in French) but it is available by request from a number of international libraries and is absolutely essential reading for this type of work in the Mediterranean. I suspect this is why Cavolinia inflexa were underrepresented and did not appear to be tied to any environmental parameter as, at this time of year, most of the population is found below the sampling depths and the samples will have just picked up the odd individual.

Rampal J. Les thécosomes (molluques pélagiques). Systématique et évolution - Écologies et biogéographie Mediterranéennes [These doctoral]. Université Aix-Marseille I1975.

Mediterranean pteropods have also been shown to undertake diurnal vertical migrations. At least two of the main species found in the nets exhibit a strong diurnal vertical migration and the sampling of each station was undertaken at different times in the day/night cycle, thus potentially skewing the data. See papers:

Andersen V, Francois F, Sardou J, Picheral M, Scotto M, Nival P. Vertical distributions of macroplankton and micronekton in the Ligurian and Tyrrhenian Seas (northwestern Mediterranean). Oceanol Acta. 1998;21(5):655–76.

Tarling GA, Matthews JBL, David P, Guerin O, Buchholz F. The swarm dynamics of northern krill (Meganyctiphanes norvegica) and pteropods (Cavolinia inflexa) during vertical migration in the Ligurian Sea observed by an acoustic Doppler current profiler. Deep-Sea Res Pt I. 2001;48(7):1671–86.

Introduction: The introduction seems to lack a depth of research, especially of older material which is still useful background on pteropod ecology. Would be good to see the original references, e.g. Lalli and Gilmer, Rampal, as well as the more modern ones, to support statements, as many of the modern studies referred to merely added

to the field of knowledge already in place.

It would be good to include some indication of why the Mediterranean is a 'climate change hotspot'.

An organism's suitability for use as a sentinel species depends on a number of factors, not only a perceived sensitivity to a climate driver, but also a sound understanding of its ecology which I am not convinced we have yet for pteropods.

Line 30: pray should be prey Note throughout: ar should be subscript when following Omega to denote saturation state of aragonite. There should be a space between the distance and the unit, e.g. 200 m Line 33-34: This is a strong statement, there are many other factors that contribute to an organism's suitability for use as a sentinel organism, make this statement more precautionary. Line 37: there is extensive work on Mediterranean pteropod distribution by Rampal which is not covered in this introduction. Lines 43-44: again, the work of Rampal should be considered here.

Methods: Line 123-125: what direction was the tow? Vertical, oblique?

It would be useful to know the depths of the stations.

Please indicate what pH scale was used.

Depth of tow is pretty shallow for many of the pteropods listed here, see Rampal 1975 for average depth distributions throughout the year. Could be that many of the species have been under-sampled.

I am not familiar with varimax rotation, please clarify for the reader.

Please explain why a CCA was chosen as opposed to another correspondence analysis, e.g. RDA? It is my understanding that CCAs are better in more controlled environments when there is a high confidence that the community has been exhaustively sampled as CCA inflates the importance of rare species. Looking at the sampling depths of used here study, I am not confident that this is the case and several common

species may have been under sampled.

Please explain why a BLRM was chosen for the analysis as opposed to, for example, a GLM?

It might help to reassure the reader that the depth or sampling time did not skew the results by including these as variables and assessing any correlation with pteropod abundance.

Results: I am not clear where the foraminifera appeared from, there is no mention of foraminifera collection in the methods or any rationale for their inclusion in the analysis prior to this point, please include in methods and some rationale for their inclusion in the introduction. Please report on the results of the BLRM in the main body of text, a p value should be reported, at least.

Some way of distinguishing whether the station numbers are located in the east or the west would be helpful.

Line 197 (and throughout): please use correct terminology to avoid confusion, this should be principle components 1 and 2.

Fig 3: this figure is very unclear and I struggle to relate the description of the figure in the text to the figure, itself. I assume that this is essentially two plots overlaid which is why there are two different axes? If not, I don't understand how the red axis relate to the PCA coordinate and the black to station coordinates as both colours cover factors 1 and 2.

Fig3 A and B: Please add to the figure description to make it clear what the coloured circles relate to, I assume that they are abundance at station and that red is east and blue is west? The overlapping of the circles renders them meaningless as it is impossible to assign the circle to the corresponding station number.

Fig 3 C: If the analysis was performed on the community as a whole, I am unclear how am unclear how C. inflexa was removed. Was it removed from just the plot or the full

analysis?

Discussion: The authors rightly point out that this study represents a "snapshot" in time and, as such, I would be wary at relating the results so strongly to ocean acidification factors and the inferring pteropod suitability as an indicator species. It is my opinion that we simply do not understand enough about pteropods to use them as an indicator species. How do the mean pteropod abundances found in this study compare to other abundance estimates from the Mediterranean? There are several timeseries that might provide a more temporally averaged estimate of abundance that could be used to validate this 'snapshot'.

Pteropod abundance is patchy and highly seasonal, we are not entirely sure what controls their lateral distribution on the water column but we do know that there are strong seasonal variations in their depth profiles (See Rampal, 1975) which may be skewing the results due to relatively shallow depth of the tow.

Comparison with foraminifera: It would be more useful to compare the pteropod abundance and distribution with a non-calcifying species to really provide some insight as to whether the differences observed are due to calcification energetics. The depths sampled are also relatively shallow for some planktonic foraminifera, e.g. O. universa, which prefer deeper waters.

Linking pteropod abundance to environmental parameters The study does not make mention of the fact that several other studies of pteropod abundance (time series) have found that pH does not have a significant effect on the abundance of pteropods through time, therefore it seem unlikely that it would have a significant effect through space. Notably, Howes et al (2015) did not find any negative effect of decreasing pH on Eastern Mediterranean pteropods. Logically, one would assume that if the effects of a gradient on pH was impacting the distributions such that pteropods were favouring the Western Mediterranean, that a decrease in the already less favourable Eastern basin would lead to a decrease in their numbers there, however this is not the case.

The authors mention that the significance of the results are being driven by the Limacinidae, but how do they explain this? I find it strange that the effects of Omega ar would be affecting one family of pteropods but not another when (as far as we know) they both have the same method of calcification and they both calcify the same polymorph of CaCO3. Looking at Fig 3C, only L. trochiformis appears to be strongly positively correlated with Omega ar, while S. subula seems to be telling the opposite story, please discuss these results. Please also include a discussion on the reasons why C. inflexa are not correlated to any variable.

Conclusions: I do not think that the results of this study can be taken to indicate that pteropods display suitability as an indicator species and I think that the assertion that they are correlated with Omega ar is an oversimplification of the results, especially when this is driven by one pteropod family and not others included in the analysis. It seems likely that the results have been skewed by sampling technique and station depth and this should be investigated before assigning the results to Omega ar.

---

## Referee Comment (RC2) · Anonymous Referee #2 · 7 Apr 2020

This paper presents an interesting dataset that was collected during a cruise across the Mediterranean Sea. The authors investigate the relationship between pteropods abundance and environmental parameters. They notably shows that the abundance of pteropods is generally higher in their study in the East Mediterranean compare to the West. They explain this difference between the two regions by the higher arago-nite saturation state in the East basin. While the data presented are interesting they suffer from severe limitations that limit the validity of the conclusions made. The main limitation is that pteropods are migratory (daily and seasonally) organisms with very patchy distribution. It is therefore virtually impossible to draw any conclusion on the relationship between pteropod abundances and environmental parameters using 22 spot measurements over thousands of km. As an example of this patchiness, their

abundance was 0 in several locations in the East Mediterranean Sea where they are supposed to be the most abundant. Furthermore, samplings were done at different time of the day at each location, which is highly problematic when studying organisms that are daily vertical migrators. I have listed below more specific comments:

L 29: Probably not the best reference

L32: Comparatively to what?

L47: Two "this" in one sentence is a bit awkward.

L54: Mediterranean Sea temperature?

L63: Why only ocean acidification?

L69-70: This sentence is not clear and true.

L72: Average by what? Yearly average? How much of this difference is due to temperature?

L123-124: Was the boat moving during the sampling? This could have big implication on what was sampled. A mesh size of 150 um is very small to capture large species. Cavolinia inflexa is for example relatively big (and can be very abundant in the Med). This could explain their low abundance.

L129: Were the samples fixed with buffered formalin? How long were the samples stored before species ID?

L144: Time of collection should be included in the PCA, this could greatly affects the presented results.

L177: Would the result of the study be the same if pteropod biomass rather than pteropod abundance was investigated?

L 235: It is not surprising to find Limacinedae as the most abundant species with this method of sampling.

L 243: If the energetic cost is driving the response of pteropod, why are they not more abundant (or at least not less abundant) in the region with more food resource?

L265: Kapsenberk paper shows data for 1 and 50 m depth.

L266-270: This sentence is not clear. What is the point made here?

L278-281: This is not true, western Mediterranean Sea temperature in winter are higher than 10. This is also not shown on Fig. 4.

L 290: Some stations are located in the Adriatic, I would not call this sea as ultra-oligotrophic.

L320: I don't see the point of comparing the data between pteropods and foraminifera here. They have been no mention of forams before in the methods and the results. This section comes from nowhere and should be deleted. I would rather like to see a comparison of the data collected here with the one collected previously in the Med Sea in other locations for example. How do those data correlate with previous data on pteropods from coastal stations or other cruises?

---

## Referee Comment (RC3) · Anonymous Referee #3 · 11 Apr 2020

The paper by Johnson et al. presents a spring distribution of environmental parameters, along with corresponding pteropod abundance and species distribution across the Mediterranean Basin. Using statistical analyses of PCA, CCA and BLMR (binary logistic model), the authors aim to delineate major environment drivers behind pteropod distribution and abundances, and compare it to foraminifera. They compare biological parameters in the W vs E of the Mediterranean basins, the split that is based on the distinct biogeochemical region. The paper amasses valuable datasets, both chemical and biological, but the current analyses and mostly interpretation raise substantial questions that need more careful addressing.

Major: 1) This study presents the distribution of omega saturation state as if this has not been published before. It is not clear if this is novel result of this study or the profiles were constructed based on data published before. If this is not clear, it is difficult to judge the suitability of carbonate chemistry data. 2) By splitting the basin into W-E before analyzing the station variability within sub-basin ignores inter- and intra-specific variability of each sub-basin. Based on the abundance data (Fig 5), there intra sub-basin variability could be as large as the W-E comparison. The same pattern is true for species distribution. As such, the authors first need to reconcile the level of variability between the stations before they can attempt to make a W-E division. 3) Based on Fig 2 and Fig 5, W and E part of the basins have comparable abundances and species distribution, the difference is really in the transition zone between the E and W (station 9, 14, 15, 16). This is the real results and not random W-E division. However, this makes a large portion of the discussion invalid and needs to be reconsidered and restructured. 4) In addition to #3, I disagree with the findings that omega is a major driver of pteropod distribution, or if it is, the co-authors need to do more throughout job to prove this. a. Given the correlation of majority of environmental parameters (temp, pH, omega, salinity; Fig 3D), it is really impossible to delineate a single driver. Contrary to Fig 5D, Fig 5C actually does not show the same co-linearity. Is this because of the exclusion of nitrate and fluorescence? The authors should attempt to present the correlation matrix of different parameters to the reader can understand the multi-collinearity. b. In the absence of one driver interpretation, I suggest that the authors stick to a multiple parameter interpretation. They delineated that various different parameters impact pteropod distribution but in the discussion they have abandoned this results and reduce it all to one, omega saturation state level. c. PCA analyses actually shows very complex relationship between environmental parameters and different pteropod species (Fig 3C), some with negative and the other species with no interaction with carbonate chemistry. However, what is baffling is the fact that PCA graphs for each species (Suppl Figure 1) actually shows that the environmental drivers are different depending on the basin. In such way, the distribution in the E part is driver by different set of parameters than the W part. In my opinion, this invalidates current analyses by subdividing the basin into E-W instead of dealing with the data on the

station level. I would strongly suggest the reiteration of the analyses on the station level to (in)validate the current results. d. Authors should analyse and interpret species distribution in greater depth based on the species habitat niches, maybe inter-specific species difference are driven by the differences in the vertical migration pattern (DVM)? How could sampling biased the results of species with deeper than 200m DVM? None of this is currently in the paper. e. Current interpretation does not hold up against the presented results and need significant restructuring. Based on the snap-shot spring distribution, the authors need to scale down the extrapolation in the discussion.

Additional: 1) There should be some background on the abundance and species distribution of pteropods in the sub-basin of the Mediterranean. That has been a well-studied topic, both spatially and temporally but authors do not include any of such data and such, fail to establish the baseline knowledge. In addition, there are no comparison of this study with the previous study on pteropods, which would give it a better comparison and evaluation. 2) The introduction fails to identify what exactly will be investigated in this paper – and that is not the population response to climate change. More structure hypothesis testing needs to be presented in the Intro. 3) How were satellite data obtained and averaged; daily/monthly? How was this done? Do you have fluorescence of chl-a data? Different figures and tables have different parameter enlisted. 4) Usually studies report the abundances in ind/m2, not m-3. Can this be, for comparison reasons, amended with 200 m vertical depth. 5) How were abundances enumerated? 6) How were the samples preserved and what was pH of the solution? 7) How do authors explain the salinity as a dominant driver, is it possible that the current distribution is water-mass related, and thus, the species have affinity to specific water mass (rather than to a environmental parameter). Please, comment on this and explain, why could you exclude the advection as a driver behind observed distribution. 8) Forams do not have a proper introduction, very confusion for a reader. 9) Figure 4 does not show Adriatic and NE part, the latitudes do not align.

---

## Author Comment (AC1) · 12 Jun 2020

Thank you for your constructive remarks. Please find our detailed responses to your comments, including expected modifications of the manuscript, below.

1. **COMMENT:** The sample depth of 200 m is relatively shallow for sampling many of the species. The early work of Rampal (1975) very clearly showed that many species exhibit seasonal depth preferences, with some populations spending many months of the year at depth in excess of 700 m. I do appreciate the difficulties in accessing this work as it is an unpublished thesis (in French) but it is available by request from a number of international libraries and is absolutely essential reading for this type of work in the Mediterranean. I suspect this is why Cavolinia inflexa were underrepresented and did not appear to be tied to any environmental parameter as, at this time of year, most of the population is found below the sampling depths and the samples will have just picked up the odd individual. Rampal J. Les thécosomes (mollugues pélagiques). Systématique et évolution - Écologies et biogéographie Mediterranéennes [These doctoral]. Université Aix-Marseille I1975.

**REPLY:** We appreciate the reviewer's comment that gave us the possibility to clarify the sampling strategy of this work. Sampling the upper 200 m water depth allowed us to capture most of the total pteropod community, though we are aware that we may have underestimated species with a deeper distribution (see below). The goal of this sampling strategy was to be able to perform a wide survey across the Mediterranean Sea during the one-month sampling research cruise.

Based on a study investigating the global distribution of pteropods, which utilised a very large dataset (25939 data points) that included 41 scientific studies (Bednaršek et al. 2012), it was found that most of the species live in the photic zone, although some pteropod species could be found at depths >500 m (Figure 1 from Bednaršek et al. 2012).

Figure 7. Pteropod carbon biomass (mg C m-3) for six depth intervals: (a) surface (0–10 m), (b) 10–25 m, (c) 25–50 m, (d) 50–200 m, (e) 200–500 m, (f)  $\geq$  500 m.

**Figure 1. From Bednaršek et al. 2012.**

From the same dataset of Bednaršek et al. (2012) in PANGAEA (doi:10.1594/PANGAEA.777387), we generated the graph in Figure 2A to show the distribution of ind. m-3 of pteropods in the upper 850 m water depth of the Mediterranean Sea. The figure only shows values from stations where the total pteropod community have been collected from below and above 200 m, to allow for a comparison between depth distributions within the water column (see map generated in Figure 2B with the station locations). These data are comprised of sampling conducted during the day and night and does not differentiate between the two. On the basis of this dataset, 93% of pteropods (individual standing stocks) are distributed in the upper 200 m (Table 1 and to be added as a supplementary table).

**Figure 2. A** Abundance of pteropods (expressed as ind. m-3) in the Mediterranean Sea from the dataset in Bednaršek et al. (2012) only including sites where pteropods have been collected below and above 200 m; **B** Location of stations within the Mediterranean Sea where data were collected above and below 200 m water depth. Unfortunately, there are a paucity of datasets in the western Mediterranean that allow us to differentiate the depth distribution of pteropods above and below 200 m (Bednaršek et al. (2012); see Figure 1 from this document, (c) depth 200-500 m and (d) below 500m for the western Mediterranean).

**Table 1.** Maximum, average, and percentage of total abundance of pteropods within the Mediterranean. A list of datasets used in this study can be found at the end of this document.

| Depth range
(m) | Max abundance
(ind. m -3 ) | Number of
observations | Avg. abundance
(ind. m -3 ) | % Abundance
(ind. m -3 ) |
|--------------------|------------------------------------------|---------------------------|-------------------------------------------|----------------------------------------|
| 0-200              | 26.67                                    | 455                       | 0.98                                      | 93.37                                  |
| 201-850            | 19                                       | 502                       | 0.07                                      | 6.63                                   |

An additional study performed in the Mediterranean South Adriatic Sea demonstrated that during a yearlong survey of the vertical distribution of gelatinous zooplankton, the large majority of pteropod species (78%) were found in the upper 50 m and 88% in the upper 200 m (Batistić et al., 2004).

We obtained a copy of Rampal's thesis (1975) that was mentioned by the Reviewer. This was not readily accessible and also (as the reviewer said) in French but we have been able to partially translate it. The data from this thesis are also not published and the author states that "The heterogeneity of the samples, in terms of collection period, sampling methodology and type of net, sampling depth, etc. does not allow one to perform a quantitative Mediterranean-wide study". We do, however, understand the value of this seminal work and we refer to it in the revised version of the manuscript accordingly.

According to Rampal (1975), Limacina bulimoides, L. trochiformis, Creseis acicula and C. conica are classified as species living in the upper 200 m water depth (epipelagic depth preference). It was suggested that *H. inflatus* and *Styliola* subula can be found at depths below 200 m (mesopelagic depth preference) (Rampal, 1975), however, recent studies show that Heliconoides inflatus primarily occurs in the upper water column (Batistić et al., 2004; Granata et al., 2020; Juranek et al., 2003) while S. subula shows greater abundance below 150 m (Andersen et al., 1998). Cavolina inflexa is classified by Rampal (1975) as a species with a bathypelagic preference, and this preference for deeper water has been corroborated by more recent studies in the Ligurian Sea focusing on this species (Granata et al., 2020; Sardou et al., 1996; Tarling et al., 2001). On the basis of Rampal's study, C. inflexa is mainly present in the western Mediterranean (mainly in the north sector) and considered rare in the eastern basin (see Figure 3 [Figure 96 from Rampal, 1975]). Rampal's thesis indicates that C. inflexa is a temperate species and, in the Mediterranean Sea, is mainly concentrated above the 40th parallel. Based on the pteropod survey of Rampal, (1975) C. inflexa is abundant only in the Catalan Sea, Gulf of Lion and Ligurian coast and in our study, only 3 of the 20 stations are included within these regions.

---

## Author Comment (AC2) · 12 Jun 2020

Thank you for your constructive remarks. Please find our detailed responses to your comments, including expected modifications of the manuscript, below.

**1.    Comment:** The main limitation is that pteropods are migratory (daily and seasonally) organisms with very patchy distribution. It is therefore virtually impossible to draw any conclusion on the relationship between pteropod abundances and environmental parameters using 22 spot measurements over thousands of km. As an example of this patchiness, their abundance was 0 in several locations in the East Mediterranean Sea where they are supposed to be the most abundant.

**2.    Comment:** Furthermore, samplings were done at different time of the day at each location, which is highly problematic when studying organisms that are daily vertical migrators.

**REPLY:** An aim of this study was to obtain an upper 200 m integrated distribution of pteropods during spring across the Mediterranean Sea's large environmental gradients in order to shed light on the potential factors modulating pteropod distribution. Sampling the upper 200 m water depth allowed us to capture most of the total pteropod community, though we are aware that we may have underestimated species with a deeper diel vertical migration and/or distribution. Patchiness is expected for zooplankton, however abundance was not 0 in several locations but only in one station (station 20 - Northern Alguero-Balear) in the western Mediterranean. The closest stations in the region also had low abundances (station 21 – 0.48 ind. m$^{-3}$ and station 22 with the second lowest abundance – 0.06 ind. m$^{-3}$).

Based on a study investigating the global distribution of pteropods, which utilised a very large dataset (25939 data points) that included 41 scientific studies (Bednaršek et al. 2012), it was found that most of the species live in the photic zone, although some pteropod species could be found at depths >500 m. From the same dataset of Bednaršek et al. (2012) in PANGAEA (doi:10.1594/PANGAEA.777387), we generated the graph in Figure 1A to show

the distribution of ind. m$^{-3}$ of pteropods in the upper 850 m water depth of the Mediterranean Sea. The figure only shows values from stations where the total pteropod community have been collected from below and above 200 m, to allow for a comparison between depth distributions within the water column (see map generated in Figure 1B with the station locations). These data are comprised of sampling conducted during the day and night and does not differentiate between the two. On the basis of this dataset, 93% of pteropods (individual standing stocks) are distributed in the upper 200 m (Table 1 and to be added as a supplementary table).

[Figure]

**Figure 1. A** Abundance of pteropods (expressed as ind. m$^{-3}$) in the Mediterranean Sea from the dataset in Bednaršek et al. (2012) only including sites where pteropods have been collected below and above 200 m; **B** Location of stations within the Mediterranean Sea where data were collected above and below 200 m water depth. Unfortunately, there are a paucity of datasets in the western Mediterranean that allow us to differentiate the depth distribution of pteropods above and below 200 m (Bednaršek et al. (2012); see Figure 1 from this document, (c) depth 200-500 m and (d) below 500m for the western Mediterranean).

**Table 1.** Maximum, average, and percentage of total abundance of pteropods within the Mediterranean. A list of datasets used in this study can be found at the end of this document.

| Depth range (m) | Max abundance (ind. m⁻³) | Number of observations | Avg. abundance (ind. m⁻³) | % Abundance (ind. m⁻³) |
|---|---|---|---|---|
| 0-200 | 26.67 | 455 | 0.98 | 93.37 |
| 201-850 | 19 | 502 | 0.07 | 6.63 |

An additional study performed in the Mediterranean South Adriatic Sea demonstrated that during a yearlong survey of the vertical distribution of gelatinous zooplankton, the large majority of pteropod species (78%) were found in the upper 50 m and 88% in the upper 200 m (Batistić et al., 2004).

On the basis of the information above, we will include in the Methods section of the revised manuscript the paragraph below, highlighting that in this study, we are characterising pteropod abundance, and that those abundances may be slightly underestimated as some species live below the sampling depth.

Added to methods section 2.2:

*Sampling the upper 200 m water depth (regardless night or day collection) will probably not capture entirely the pteropod community at each location, however, it allows to quantify most of the total pteropod abundance. Based on a study investigating the global distribution of pteropods, which utilised a very large dataset (25939 data points) that included 41 scientific studies (Bednaršek et al. 2012), it was found that most of the species live in the photic zone and that in the Mediterranean Sea, specifically, pteropod abundance below 200 m is more than one order of magnitude lower (mean $0.07 \pm 0.89$ ind. m⁻³) than the abundance in the upper 200 m (mean $1.04 \pm 2.77$ ind. m⁻³) with 93% and 7% of pteropods distributed above and below the 200 m, respectively (Table 1 here and added as a supplementary table in the revised manuscript).*

We understand that *Cavolina inflexa*, as well as *Styliola subula*, have a depth preference below 200 m and a strong diel and seasonal variations in their depth

distribution habitat (Andersen et al., 1998; Rampal, 1975; Tarling et al., 2001). For this reason, we have decided to be more conservative in our approach and not incorporate *S. subula* and *C. inflexa* into the discussion of pteropod ecology and into our statistical analysis. We will focus on the species that mainly live in the upper 200 m. As these species formed a very small portion of the total abundance, when we compare the original Canonical Correspondence Analysis (CCA) against the new CCA (where some variables were removed to avoid collinearity), the alignment of species with the significant environmental parameters (temperature and $\Omega_{ar}$) is unchanged. The results and significance for the Pearson's correlations are also unchanged (please see response to comments 16, 17, 18 and 19 for details on the updated statistical analyses).

We have added the following paragraph in the revised Methods section to clarify our approach:

*We are aware that vertical distribution of pteropods is species-specific.* Limacina bulimoides, L. trochiformis, C. acicula *and* C. conica *are classified as surface and subsurface species (Rampal, 1975).* Heliconoides inflatus *and* S. subula *can be found at depths below 200 m and are classified as mesopelagic by Rampal (1975), however recent studies show that* H. inflatus *primarily occurs in the upper water column (Batistić et al., 2004; Granata et al., 2020; Juranek et al., 2003) while* S. subula *shows the greatest abundance below 150 m* (Andersen et al., 1998)*.* C. inflexa *is classified by Rampal (1975) as a bathypelagic species with a distribution extending below 1000 m, and this preference for deeper water has been corroborated by more recent studies in the Ligurian Sea focusing on this species (Granata et al., 2020; Sardou et al., 1996; Tarling et al., 2001). Due to the strong diel and seasonal variations in their depth distribution habitat (Andersen et al., 1998; Rampal, 1975; Tarling et al., 2001), we decided not to incorporate* S. subula *and* C. inflexa *into our statistical analyses of distribution in relation to the environmental parameters.*

The collection of day and night samples at each location was not anticipated in our research. However, we considered diurnal vertical migrations (visual interpretation, ANOVAs using a categorical approach by dividing stations

between day and night) and found no significant difference in total or species abundances when comparing results between day and night sampling (Table 2). We also sampled 10 stations during the day and 10 during the night, spread across the Mediterranean. For a multivariate approach, we conducted a CCA which indicated that PAR (photosynthetically active radiation) did not heavily affect pteropod community composition within the Mediterranean Sea (Figure 2).

As stated in the original manuscript – "After an initial analysis, PAR (photosynthetically active radiation) was removed as it did not significantly contribute to the variation of environmental parameters." We acknowledge that species do undertake diurnal migration and we have made a point in the revised manuscript that the time of day for collection did not appear to be a driving factor affecting abundances in the upper 200 m integrated samples of this study. Updated text:

*Many pteropod species undertake diurnal vertical migration, and after an initial analysis, PAR was removed as it did not appear to be a driving factor affecting pteropod's abundances in the upper 200 m integrated samples of this study (Supplementary Figure – CCA and ANOVA using day/night stations).*

**Table 2.** ANOVA table using night/day as the dependent variable against total and species abundances.

| | | Sum of Squares | df | Mean Square | F | Sig. |
|---|---|---|---|---|---|---|
| total abundance | Between Groups | 1.988 | 1 | 1.988 | .713 | .409 |
| | Within Groups | 50.182 | 18 | 2.788 | | |
| | Total | 52.170 | 19 | | | |
| *H. inflatus* | Between Groups | .000 | 1 | .000 | .000 | .987 |
| | Within Groups | 4.317 | 18 | .240 | | |
| | Total | 4.317 | 19 | | | |

| | | | | | | |
|---|---|---|---|---|---|---|
| *L. trochiformis* | Between Groups | .354 | 1 | .354 | 1.029 | .324 |
| | Within Groups | 6.198 | 18 | .344 | | |
| | Total | 6.552 | 19 | | | |
| *L. bulimoides* | Between Groups | .251 | 1 | .251 | .789 | .386 |
| | Within Groups | 5.726 | 18 | .318 | | |
| | Total | 5.977 | 19 | | | |
| *C. acicula* | Between Groups | .002 | 1 | .002 | .056 | .816 |
| | Within Groups | .590 | 18 | .033 | | |
| | Total | .592 | 19 | | | |
| *C. conica* | Between Groups | .003 | 1 | .003 | .121 | .732 |
| | Within Groups | .400 | 18 | .022 | | |
| | Total | .403 | 19 | | | |

**Axis 1 and 2 (63.6 %)**

[Figure]

**Figure 2**. Triplot from the Canonical Correspondence Analysis (including the PAR variable), where the relation between species and environmental variables are

obtained from the proximity (or remoteness) of species labels and environmental arrows (tips). The length of the arrow represents the importance (contribution) of the environmental variable to the data structure, thus we can see that the PAR variable, due to its short arrow, does not heavily contribute to pteropod community composition within the Mediterranean.

3.    **Comment:** L 29: Probably not the best reference

**REPLY:** Thank you. This reference has been changed to Fabry (1989).

4.    **Comment:** L32: Comparatively to what?

**REPLY:** This line has been changed to:

*Pteropods are very susceptible to changes in carbonate saturation state (Ω) due to their aragonite shell, which is a more soluble form of calcium carbonate compared to other morphs such as calcite (Mucci et al., 1989).*

5.    **Comment:** L47: Two "this "in one sentence is a bit awkward.

**REPLY:** This line has been removed.

6.    **Comment:** L54: Mediterranean Sea temperature?

**REPLY:** This line has been updated:

Mediterranean Sea surface temperature is expected to rise by 1.5-2°C by the end of this century, with atmospheric warming likely to be 20% faster than the global average (Lazzari et al., 2013; Lionello and Scarascia, 2018).

7.    **Comment:** L63: Why only ocean acidification?

**REPLY:** This line does not only refer to ocean acidification but the changes associated with ocean warming and acidification - "how organisms and communities will respond to ocean conditions under climate change."

**8.    Comment:** L69-70: This sentence is not clear and true.

**REPLY:** Thank you for your recommendation. This sentence has been updated:

*The Mediterranean Sea is a semi-enclosed evaporitic basin characterised by an overall lower sea surface temperature (SST) and salinity in the western basin compared to the eastern basin. This is a consequence of its anti-estuarine circulation with surface Atlantic water entering through the Gibraltar Strait being modified moving eastward (Rohling et al., 2009).*

**9.    Comment:** L72: Average by what? Yearly average?

**REPLY:** Thank you for indicating the need for clarification.  This sentence will be updated as:

*Using data collected from the MedSeA research cruise (2013)* (D'Amario et al., 2017a, 2018; Gemayel et al., 2015; Hassoun et al., 2015a, 2015b; Mallo et al., 2017) *the average $\Omega_{ar}$ (saturation state of aragonite) in the upper 200 m water depth gradually increased from approximately 2.7 in the Atlantic to approximately 3.6 in the Eastern Mediterranean.*

**10.    Comment:** L123-124: Was the boat moving during the sampling? This could have big implication on what was sampled.

**REPLY:** The methods section has been updated with:

*The plankton towing was performed with the vessel moving at approximately 1 nautical knot, with an oblique sampling direction.*

**11.  COMMENT:** A mesh size of 150 um is very small to capture large species. Cavolinia inflexa is for example relatively big (and can be very abundant in the Med). This could explain their low abundance.

**REPLY:** The effect of net and mesh size on global pteropod biomass was investigated in Bednaršek et al., (2012) and the results indicated that average biomass was similar, regardless of the size of mesh that was used in sampling. Wells et al. (1975) and most recently Howes et al. (2014) highlighted that a larger mesh size (e.g. 330 μm) can result in underestimating (the often more abundant) pteropod juveniles and small-sized adult species (such as Limacinadae). A large mesh size will also result in the collection of other large-sized zooplankton, which can result in damaging the fragile pteropod shell, making identification and counting much more challenging.

Further, the abundance ranges in this study are comparable to other studies in the Mediterranean when using a variety of mesh sizes, with the exception of very coastal sites (Table 3 in comment #22).

On the basis of the reasons above, we believe that the mesh size used in our study provides a good estimate of the pteropod community in the upper 200 m.

Please refer to the reply in comment #2 regarding our strategy with *C. inflexa*.

Updated text:

*The mesh size in this study gives a good estimate of the pteropod community, aimed at including juveniles, the adults of small species (Howes et al., 2014), and large adults. Although some large adult individuals may have escaped collection, the majority of the pteropod community could be sampled. The effect of net and mesh size on global pteropod biomass was investigated in Bednaršek et al., (2012) and the results indicated that average biomass was similar, regardless of the size of mesh that was used in sampling.*

**12.     Comment:** L129: Were the samples fixed with buffered formalin? How long were the samples stored before species ID?

**REPLY:** The Methods have been updated with:

*Plankton samples were preserved on board in a 4% formaldehyde solution that was buffered with Hexamethylenetetramine at pH 8.2 and were stored in 500 ml polycarbonate bottles at 4ºC in the dark. pH was measured in all the samples, at the beginning, during and the end of the storing period to ensure that the state of the pteropod shells were not affected by the preservation technique. The samples were processed a few weeks after the collection.*

**13.     Comment:** L144: Time of collection should be included in the PCA, this could greatly affects the presented results.

**REPLY:** Please refer to our reply to comments 1 and 2 regarding time of collection.

**14.     Comment:** L177: Would the result of the study be the same if pteropod biomass rather than pteropod abundance was investigated?

**REPLY:** Detailed biomass data are not the focus of this paper and they are in the process of being analysed (length and weight data). Biomass is a function of size and abundance. Where there are no weight or length data available for pteropods, biomass is usually calculated using abundance and general length-weight conversions for particular species or groups, as in global study of pteropod biomass by Bednaršek et al., (2012). Most of the organisms in this study were juveniles and the adults were rather evenly distributed between the stations. We anticipate that the trend will be similar whether biomass or abundance was investigated. We are basing the results in this paper on count data only.

**15.     Comment:** L 235: It is not surprising to find Limacinedae as the most abundant species with this method of sampling.

**REPLY:** It is not clear what specific part of the sampling strategy the reviewer is referring to. For instance, Manno et al. (2019; Tyrrhenian Sea) and Fernandes de Puelles et al. (2007; Balearic Sea) found Creseiidae to be the dominant family, despite using a mesh size similar to ours (200 and 100-120 μm, respectively). All sampling methods contain an element of bias and there is no one way to determine the absolute abundance of all species. This was also noted in Howes et al. (2015) that suggested an undersampling of juvenile *Heliconoides inflatus* due to using a 330 μm mesh size. We have commented on mesh size in the Methods section of the revised manuscript as a potential sampling bias (please see our response to comment #11).

**16.    Comment:** L 243: If the energetic cost is driving the response of pteropod, why are they not more abundant (or at least not less abundant) in the region with more food resource?

**REPLY:** This was explained in the manuscript in the section **Sensitivity to $\Omega_{ar}$,** however, we will further clarify this point and rephrase this section:

*Pteropods need to split the energetic cost between calcification and catching food. The energetic cost of calcification is generally very high (Sanders et al., 2018). Conversely, pteropods can starve for long periods (Busch et al., 2014). In pteropods, shell calcification is important for balance and defence (Harbison and Gilmer, 1992). Further Watson et al. (2017) showed that an increase in the cost of carbonate deposition, including from the projected decrease in pH, may lead to a ~50 to 70% increase in the proportion of the total energy budget required for shell production, to a doubling of the $CaCO_3$ deposition cost. Changes in the energy budget allocation to shell cost would likely alter ecological trade-offs between calcification and other drivers (Watson et al., 2017). What we suggest in the paper is that high $\Omega_{ar}$ is likely more important than a high nutrient environment. If there is not a strong $\Omega_{ar}$ gradient, it is likely that pteropods will follow a food gradient, as in Burridge et al. (2017).*

**17.    Comment:** L265: Kapsenberk paper shows data for 1 and 50 m depth.

**REPLY:** Thank you for pointing this out. This will be updated in text: locations (~15 m in the Gulf of Trieste and 1 and 50 m in Villefranche-sur-Mer).

**18.    Comment:** L266-270: This sentence is not clear. What is the point made here?

**REPLY:** This sentence has been removed as it uses references that are no longer required for the point regarding seasonality of $\Omega_{ar}$ saturation in the Mediterranean Sea.

**19.    Comment:** L278-281: This is not true, western Mediterranean Sea temperature in winter are higher than 10. This is also not shown on Fig. 4.

**REPLY:** These are values reported in Rohling et al. (2009). Figure 4B shows the gradient of temperature in the Mediterranean from the data collected on our cruise, and the text will be updated to indicate this.

*In the Mediterranean, sea surface temperatures (SST) vary by about 10°C over the year, with the north-western Mediterranean having a winter average of approximately 10°C and a summer average of 21°C, while the south-eastern Mediterranean winter average is approximately 15°C and the summer approximately 26°C (Naval Oceanography Command, 1987, as cited in Rohling et al., 2009), with a consistent west to east gradient as illustrated in Fig. 4B (from data collected on the MedSeA research cruise, 2013).*

**20.    Comment:** L 290: Some stations are located in the Adriatic, I would not call this sea as ultraoligotrophic.

**REPLY:** Thank you for pointing this out. The sentence has been updated:

*In our study, pteropod abundance was ~5x greater in the largely ultra-oligotrophic eastern Mediterranean (not including the Adriatic Sea).*

**21.   Comment:** L320: I don't see the point of comparing the data between pteropods and foraminifera here. They have been no mention of forams before in the methods and the results. This section comes from nowhere and should be deleted.

**REPLY:** This paper is providing the first comparison of pteropod and foraminifera distribution in the Mediterranean Sea. Foraminifera are single-celled marine eukaryotes producing calcite and little is known about how the distribution of these important groups of calcifying zooplankton differs. The foraminifera were collected in the same plankton towing samples as the pteropods of this study and the results were published in BG (https://www.biogeosciences.net/14/2245/2017/), adding value to this work.

As mentioned, the published foraminifera study used data collected from the same set of samples and has been already published. This was mentioned in the introduction and we refer to Mallo et al. (2017) for details on the methodology. We realise that for clarity this information could be expanded on in order to provide a better rationale for their inclusion. We added the following text in the Introduction, Methods and Results sections.

Introduction:

*We also present the relationship between pteropods distribution and another major group of planktic marine calcifier, foraminifera (single-celled, calcareous zooplankton). Investigating this relationship between pteropods and foraminifera is important as ocean acidification has been shown to cause ecosystem shifts due to altered competition between calcareous species, likely resulting from the physiological responses of individual species (Kroeker et al., 2013). Foraminifera were collected during the same research cruise campaign (data published in Mallo et al., 2017) and are therefore directly comparable with this study on thecosome pteropods, giving us the opportunity to investigate the ecological relationship between groups of prime calcifying zooplankton in the Mediterranean Sea.*

Methods:

*Samples for foraminifera were collected in the same net as the pteropod samples at all of the same stations and preserved using the same methodology. Foraminifera were identified to species level using a light microscope and following the guidelines and taxonomic nomenclature of André et al. (2013), Aurahs et al. (2011), Hemleben et al. (1989) and Spezzaferri et al. (2015), depending upon each species. For a more detailed description of collection, preservation and taxonomic identification methods, please refer to Mallo et al. (2017).*

Statistical Methods:

*Planktic foraminifera total abundance and distribution presented in Mallo et al. (2017) were compared to the pteropod data from this study. The tow samples from Mallo et al. (2017) were collected during the same cruise and within the same nets as the pteropods of the present study, allowing a direct comparison of these two groups of key planktic calcifiers. To compare the abundance of pteropods and foraminifera within specific regions of the Mediterranean Basin, we used a Generalised Linear Mixed Model (GLMM) with a gamma distribution. As the magnitude of the abundance data is very different between pteropods and foraminifera (almost one order of magnitude), the abundance data was transformed to its logarithmic scale to make abundances from both groups comparable. For this analysis, the Mediterranean was split into two basins: "western" stations (1, 2, 3, 5, 6, 7, 19, 20, 21, 22) and "eastern" stations (11, 12, 13, 14, 15, 16, 17, 16-18).*

*To run the GLMM, the functions "glm" in the glmmTMB package was used (Brooks et al., 2017).*

Results:

*The results from the GLMM comparing the aggregated abundance of pteropods and foraminifera between the two basins (Eastern and Western basins), indicates*

*that there are significant differences between the abundance of both taxa (chisq = 29.27, p < 0.05), between the Eastern and Western Mediterranean basins (chisq = 5.57, p < 0.05), and also in their interaction (chisq = 4.97, p < 0.05). These results indicates that an inverse relationship between taxa abundance and Mediterranean basin exist.*

*Pteropod abundance is distinctly greater in the Eastern ($\bar{x}$=2.13 ind. m$^{-3}$) Mediterranean than in the Western Mediterranean ($\bar{x}$=0.47 ind. m$^{-3}$), while foraminiferal populations showed a contrasting abundance (Figure 3 here and to be added to the revised manuscript) with higher abundance in the Western Mediterranean ($\bar{x}$=1.87 ind. 10 m$^{-3}$) than in the Eastern Mediterranean ($\bar{x}$=0.96 ind. 10 m$^{-3}$).*

[Figure]

**Figure 3.** Box plot showing the contrasting abundance distribution between pteropods and foraminifera between the East and West of the Mediterranean.

**22.** **Comment:** I would rather like to see a comparison of the data collected here with the one collected previously in the Med Sea in other locations for

example. How do those data correlate with previous data on pteropods from coastal stations or other cruises?

**REPLY:** An additional section on pteropods in the Mediterranean has been added to the Discussion to give more context to our results. Table 2 compiles relevant published pteropod studies in the Mediterranean Sea and is included here and in the supplementary material of the revised manuscript.

*This study aims to give the most comprehensive pteropod distribution within the upper 200 m across the Mediterranean Sea during the spring period. A previous investigation of the whole basin was made by Rampal (1975) who performed a comparative analyses of pteropod abundance within the different Mediterranean sectors. Unfortunately, the heterogeneity of the collected materials limited the quantitative approach of this study and the results are not presented in terms of pteropod concentration. The only published Mediterranean long-term study focusing solely on pteropods (Howes et al., 2015) is in the Ligurian Sea (water depth 0-70m depth). In this study, the dominant species in each family were* C. acicula *and* H. inflatus, *corroborating well with our overall findings. Contrary to our results, in this study, Limacinadae was found to be the least abundant family, which the authors contributed to a sampling bias which led to under-sampling. Abundance average of pteropods in Howes et al. (2015) over the period from 1957-2003 was 15.7 ind. m$^{-3}$ for family Creseidae, and 5.5 ind. m$^{-3}$ for family Limacinadae (this includes* H. inflatus*). Other studies investigating pteropod community abundance in different Mediterranean Sea regions (e.g. Ligurian Sea, Balearic Sea, Adriatic Sea, Tyrrhenian Sea) show average abundances of the entire pteropod community ranging from 0.34 – 5.9 ind. m$^{-3}$ (Batistić et al., 2004; Fernández de Puelles et al., 2007; Granata et al., 2020; Manno et al, 2019; Table 3 here and added as a supplementary Table to the revised manuscript). Pteropod abundance in our study is as high as 5.14 ind. m$^{-3}$ and within the same magnitude as most of these other Mediterranean studies, with an average of 1.22 ind. m$^{-3}$ across the whole Mediterranean. In agreement with our results, Granata et al. 2020 and Batistić et al., 2004 also found* H. inflatus *to be the most abundant pteropod species. However, all the previous studies mentioned above differed in*

*sampling methodology, were sampled over different seasons and time periods, and are coastal versus open sea stations, making a direct comparison between the studies and regions difficult.*

**Table 3.** An overview of published pteropod studies in the Mediterranean

| Region of Collection | Min-max conc. of pteropods community (ind. m$^{-3}$) | x̄ conc. (ind. m$^{-3}$) | Period of sampling | Collection depth | Water column depth (m) | Most abundant species/taxa | Net/mesh size | Reference |
|---|---|---|---|---|---|---|---|---|
| Ligurian and Tyrrhenian Seas (NW Mediterranean) | Study focuses on *C. inflexa, C. pyramidata* and *S. subula*).

Min-max not provided | Day:
*Cavolina inflexa*: 4.0
*Clio pyramidata*: 2.1
*Styliola subula*: 0.3
Night:
*Cavolina inflexa*: 1.7
*Clio pyramidata*: 1.6
*Styliola subula*: 0.4 | April, 1994 | 0-25
25-50
50-75
0-75
75-150
100-150
150-200
150-250
250-350
350-400
400-450
450-500
500-550
550-700 | Various 700-2700 | *Cavolina inflexa* | BIONESS 1 m$^2$ mouth 500 µm mesh | Andersen et al., 1998 |
| Southern Adriatic | Min: 0.38
Max: 57.68 | 2.87 | April September November February June | 0–50
50–100
100–200
200–300
300–400
400–600
600–1000 | 1242 | *Heliconoides inflatus* | 113 cm diameter 380 cm length 250 µm mesh | Batistić et al., 2004 |
| Balearic Sea | Only monthly x̄ given.
Min: 4
Max: 11 | 5.9 | 1994-2003 (all year round) | 0-75
0-100 | Various 78-200 | *Creseis acicula* | Bongo-20 Plankton net 100 µm and 120 µm meshes | Fernández de Puelles et al., 2007 |
| Ligurian Sea | *C. inflexa:* 1.59 (20-40 m)
*C. pyramidata:* 0.06 (100- 200 m)
*H. inflatus:* 6.87(0-20 m) | 0.3 | April-May, 2013 | 0-20
20-40
40-60
60-80
80-100
100-200
200-400
400-600
600-800
800-1000
1000-1300 | Various 1400-1639 | *Heliconoides inflatus* | BIONESS multinet 1 m$^2$ mouth 230 µm mesh | Granata et al., 2020 |
| NW Ligurian Sea | Creseidae: ~630
Cavoliniidae: ~790
Limacinadae (incl. *H. inflatus*): max 60.8 | Creseidae: 15.7
Cavoliniidae: 13.8
Limacinadae: 5.5 | 1967-2003 (all year round) | 0–75 | ~80 | Creseidae | Juday Bogorov net 330 µm mesh 50 cm diameter | Howes et al. (2015) |
| Tyrrhenian Sea | Min: 0.00
Max: 4.02 | *C. acicula*: 1.48 | August, 2015 | One depth for each station according to | Various (73-185) | *Creseis acicula* | Bongo-40 200 µm mesh | (Manno et al., 2019) |

| | | C. conica: 1.11  H. inflatus: 1.03  L. trochiformis: 0.64  L. bulimoides: 0.33 | | | the sea bottom (min depth 0-65, Max depth 0-170) | | | | |
|---|---|---|---|---|---|---|---|---|---|
| Eastern Mediterranean | Sicilian Channel: Max. 120 ind. m$^{-3}$ | Sicily Channel: 2.07  Ionion Sea: 0.56  Cretan Sea: 1.00  Cretan Passage: 2.32  Rhodes area: 2.94  Levantine Sea: 1.37 | | October-November | 300 | Various: 449-4359 | N/A | WP-3 net 113 cm diameter 200 μm mesh | Mazzocchi et al., 1996 |
| Ligurian Sea | *Study focuses solely on Cavolina inflexa*  Max: x̄ 1.64 (0-200 m) | Not provided | | September, 1997 | 0-500 with intervals 0-25 25-50 50-75 100-125 125-150 150-200 | Not provided | *Cavolina inflexa* | MOCNESS 1 m$^2$ mouth 300 μm and 2000 μm meshes | Tarling et al., 2001 |

**REFERENCES**

André, A., Weiner, A., Quillévéré, F., Aurahs, R., Morard, R., Douady, C. J., de Garidel-Thoron, T., Escarguel, G., de Vargas, C. and Kucera, M.: The cryptic and the apparent reversed: lack of genetic differentiation within the morphologically diverse plexus of the planktonic foraminifer *Globigerinoides sacculifer* , Paleobiology, 39(1), 21–39, doi:10.1666/0094-8373-39.1.21, 2013.

Aurahs, R., Treis, Y., Darling, K. and Kucera, M.: A revised taxonomic and phylogenetic concept for the planktonic foraminifer species *Globigerinoides ruber* based on molecular and morphometric evidence, Mar. Micropaleontol., 79(1–2), 1–14, doi:10.1016/j.marmicro.2010.12.001, 2011.

Batistić, M., Kršinić, F., Jasprica, N., Carić, M., Viličić, D. and Lučić, D.: Gelatinous invertebrate zooplankton of the South Adriatic: Species composition and vertical distribution, J. Plankton Res., 26(4), 459–474, doi:10.1093/plankt/fbh043, 2004.

Bednaršek, N., Možina, J., Vogt, M., Brien, C., Tarling, G. A., Mozina, J., Vogt,

M., O'Brien, C. and Tarling, G. A.: The global distribution of pteropods and their contribution to carbonate and carbon biomass in the modern ocean, Earth Syst. Sci. Data, 4(1), 167–186, doi:10.5194/essd-4-167-2012, 2012.

Brooks, M. E., Kristensen, K., Benthem, K. J. van, Magnusson, A., Berg, C. W., Nielsen, A., Skaug, H. J., Maechler, M., Bolker, B. M., van Benthem, K. J., Magnusson, A., Berg, C. W., Nielsen, A., Skaug, H. J., Mächler, M. and Bolker, B. M.: {glmmTMB} Balances Speed and Flexibility Among Packages for Zero-inflated Generalized Linear Mixed Modeling, R J., 9(2), 378–400, doi:10.32614/rj-2017-066, 2017.

Burridge, A. K., Goetze, E., Wall-Palmer, D., Le Double, S. L., Huisman, J. and Peijnenburg, K. T. C. A.: Diversity and abundance of pteropods and heteropods along a latitudinal gradient across the Atlantic Ocean, Prog. Oceanogr., 158, 213–223, doi:10.1016/j.pocean.2016.10.001, 2017.

Busch, D. S., Maher, M., Thibodeau, P. and McElhany, P.: Shell Condition and Survival of Puget Sound Pteropods Are Impaired by Ocean Acidification Conditions, edited by G. E. Hofmann, PLoS One, 9(8), e105884, doi:10.1371/journal.pone.0105884, 2014.

D'Amario, B., Ziveri, P., Grelaud, M., Oviedo, A. and Kralj, M.: Coccolithophore haploid and diploid distribution patterns in the Mediterranean Sea: can a haplo-diploid life cycle be advantageous under climate change?, J. Plankton Res., 39(5), 781–794, doi:10.1093/plankt/fbx044, 2017.

D'Amario, B., Ziveri, P., Grelaud, M. and Oviedo, A.: *Emiliania huxleyi* coccolith calcite mass modulation by morphological changes and ecology in the Mediterranean Sea, edited by S. Rutherford, PLoS One, 13(7), e0201161, doi:10.1371/journal.pone.0201161, 2018.

Fabry, V. J.: Aragonite production by pteropod molluscs in the subarctic Pacific, Deep Sea Res. Part A, Oceanogr. Res. Pap., 36(11), 1735–1751, doi:10.1016/0198-0149(89)90069-1, 1989.

Fernández de Puelles, M. L., Alemany, F. and Jansá, J.: Zooplankton time-series in the Balearic Sea (Western Mediterranean): Variability during the decade 1994–2003, Prog. Oceanogr., 74(2–3), 329–354, doi:10.1016/j.pocean.2007.04.009, 2007.

Gemayel, E., R Hassoun, A. E., Benallal, M. A., Goyet, C., Rivaro, P., Abboud-Abi Saab, M., Krasakopoulou, E., Touratier, F., Ziveri, P., Hassoun, A. E. R., Benallal, M. A., Goyet, C., Rivaro, P., Abboud-Abi Saab, M., Krasakopoulou, E., Touratier, F. and Ziveri, P.: Climatological variations of total alkalinity and total dissolved inorganic carbon in the Mediterranean Sea surface waters, Earth Syst. Dyn., 6(2), 789–800, doi:10.5194/esd-6-789-2015, 2015.

Granata, A., Bergamasco, A., Battaglia, P., Milisenda, G., Pansera, M., Bonanzinga, V., Arena, G., Andaloro, F., Giacobbe, S., Greco, S., Guglielmo, R., Spanò, N., Zagami, G. and Guglielmo, L.: Vertical distribution and diel migration of zooplankton and micronekton in Polcevera submarine canyon of the Ligurian mesopelagic zone (NW Mediterranean Sea), Prog. Oceanogr., 183, 102298, doi:https://doi.org/10.1016/j.pocean.2020.102298, 2020.

Harbison, G. R. and Gilmer, R. W.: Swimming, buoyancy and feeding in shelled pteropods: a comparison of field and laboratory observations, J. Molluscan Stud., 58(3), 337–339, doi:10.1093/mollus/58.3.337, 1992.

Hassoun, A. E. R., Gemayel, E., Krasakopoulou, E., Goyet, C., Abboud-Abi Saab, M., Guglielmi, V., Touratier, F. and Falco, C.: Acidification of the Mediterranean Sea from anthropogenic carbon penetration, Deep Sea Res. Part I Oceanogr. Res. Pap., 102, 1–15, doi:10.1016/J.DSR.2015.04.005, 2015a.

Hassoun, A. E. R., Guglielmi, V., Gemayel, E., Goyet, C., Abboud-Abi Saab, M., Giani, M., Ziveri, P., Ingrosso, G. and Touratier, F.: Is the Mediterranean Sea Circulation in a Steady State, J. Water Resour. Ocean Sci., 4(1), 6, doi:10.11648/j.wros.20150401.12, 2015b.

Hemleben, C., Spindler, M., Anderson, O. R., Hemleben, C., Spindler, M. and Anderson, O. R.: Taxonomy and Species Features, in Modern Planktonic Foraminifera, pp. 8–32, Springer New York., 1989.

Kroeker, K. J., Kordas, R. L., Crim, R., Hendriks, I. E., Ramajo, L., Singh, G. S., Duarte, C. M. and Gattuso, J.-P.: Impacts of ocean acidification on marine organisms: quantifying sensitivities and interaction with warming, Glob. Chang. Biol., 19(6), 1884–1896, doi:10.1111/gcb.12179, 2013.

Lazzari, P., Mattia, G., Solidoro, C., Salon, S., Crise, A., Zavatarelli, M., Oddo, P. and Vichi, M.: The impacts of climate change and environmental management policies on the trophic regimes in the Mediterranean Sea: Scenario analyses, J. Mar. Syst., 135, 137–149, doi:10.1016/j.jmarsys.2013.06.005, 2013.

Lionello, P. and Scarascia, L.: The relation between climate change in the Mediterranean region and global warming, Reg. Environ. Chang., 18(5), 1481–1493, doi:10.1007/s10113-018-1290-1, 2018.

Mallo, M., Ziveri, P., Mortyn, P. G., Schiebel, R. and Grelaud, M.: Low planktic foraminiferal diversity and abundance observed in a 2013 West-East Mediterranean Sea transect, Biogeosciences Discuss., 1–31, doi:10.5194/bg-2016-266, 2017.

Manno, C., Rumolo, P., Barra, M., d'Albero, S., Basilone, G., Genovese, S., Mazzola, S. and Bonanno, A.: Condition of pteropod shells near a volcanic $CO_2$ vent region, Mar. Environ. Res., 143, 39–48, doi:10.1016/j.marenvres.2018.11.003, 2019.

Mazzocchi, M., Christou, E., Fragopoulu, N. and Siokoufrangou, I.: Mesozooplankton distribution from Sicily to Cyprus (Eastern Mediterranean) .1. General aspects, Oceanol. Acta, 20(3), 521–535, 1996.

Mucci, A., Canuel, R. and Zhong, S.: The solubility of calcite and aragonite in

sulfate-free seawater and the seeded growth kinetics and composition of the precipitates at 25°C, Chem. Geol., 74(3–4), 309–320, doi:10.1016/0009-2541(89)90040-5, 1989.

Rohling, E. J., Abu-Zied, R., Casford, J. S. L., Hayes, A. and Hoogakker, B.: The Marine Environment: Present and Past, in The Physical Geography of the Mediterranean, pp. 33–67., 2009.

Sanders, T., Schmittmann, L., Nascimento-Schulze, J. C. and Melzner, F.: High Calcification Costs Limit Mussel Growth at Low Salinity, Front. Mar. Sci., 5(OCT), 352, doi:10.3389/fmars.2018.00352, 2018.

Skjoldal, H. R., Wiebe, P. H., Postel, L., Knutsen, T., Kaartvedt, S. and Sameoto, D. D.: Intercomparison of zooplankton (net) sampling systems: Results from the ICES/GLOBEC sea-going workshop, Prog. Oceanogr., 108, 1–42, doi:https://doi.org/10.1016/j.pocean.2012.10.006, 2013.

Spezzaferri, S., Kucera, M., Pearson, P. N., Wade, B. S., Rappo, S., Poole, C. R., Morard, R. and Stalder, C.: Fossil and Genetic Evidence for the Polyphyletic Nature of the Planktonic Foraminifera "*Globigerinoides*", and Description of the New Genus Trilobatus, edited by S. Abramovich, PLoS One, 10(5), e0128108, doi:10.1371/journal.pone.0128108, 2015.

Tarling, G. A., Matthews, J. B. L., David, P., Guerin, O. and Buchholz, F.: The swarm dynamics of northern krill (Meganyctiphanes norvegica) and pteropods (*Cavolinia inflexa*) during vertical migration in the Ligurian Sea observed by an acoustic Doppler current profiler, Deep. Res. Part I Oceanogr. Res. Pap., 48(7), 1671–1686, doi:10.1016/S0967-0637(00)00105-9, 2001.

Watson, S. A., Morley, S. A. and Peck, L. S.: Latitudinal trends in shell production cost from the tropics to the poles, Sci. Adv., 3(9), doi:10.1126/sciadv.1701362, 2017.

Wells, F. E.: Effects of mesh size on estimation of population densities of

tropical euthecosomatous pteropods, Mar. Biol., 20(4), 347–350, doi:10.1007/BF00354276, 1973.

**Datasets used in Table 1**

Koppelmann, Rolf; Weikert, Horst (2008): Plankton abundance of mocness net M44/4_D-MOC216. doi:10.1594/PANGAEA.249974

Koppelmann, Rolf; Weikert, Horst (2008): Plankton abundance of mocness net M44/4_D-MOC220. doi:10.1594/PANGAEA.249975

Koppelmann, Rolf; Weikert, Horst (2008): Plankton abundance of mocness net M44/4_D-MOC242. doi:10.1594/PANGAEA.249977

Koppelmann, Rolf; Weikert, Horst (2008): Plankton abundance of mocness net M44/4_D-MOC249. doi:10.1594/PANGAEA.249978

Koppelmann, Rolf; Weikert, Horst (2008): Plankton abundance of mocness net M44/4_D-MOC268. doi:10.1594/PANGAEA.81995

Koppelmann, Rolf; Weikert, Horst (2008): Plankton abundance of mocness net M44/4_D-MOC273. doi:10.1594/PANGAEA.81998

Koppelmann, Rolf; Weikert, Horst (2008): Plankton abundance of mocness net M44/4_D-MOC280. doi:10.1594/PANGAEA.81999

Koppelmann, Rolf; Weikert, Horst (2008): Plankton abundance of mocness net M44/4_D-MOC281. doi:10.1594/PANGAEA.249979

Koppelmann, Rolf; Weikert, Horst (2008): Plankton abundance of mocness net M44/4_D-MOC293. doi:10.1594/PANGAEA.249980

Koppelmann, Rolf; Weikert, Horst (2008): Plankton abundance of mocness net M44/4_D-MOC304. doi:10.1594/PANGAEA.249982

Mazzocchi, Maria Grazia (2008): Mesozooplankton abundance and species composition in the Ionian Sea in April-May 1992. Part 2. Stazione Zoologica Anton Dohrn, doi:10.1594/PANGAEA.703258

Mazzocchi, Maria Grazia (2008): Mesozooplankton abundance and species composition in the Ionian Sea in April-May 1999. Stazione Zoologica Anton Dohrn, doi:10.1594/PANGAEA.703201

Mazzocchi, Maria Grazia (2008): Mesozooplankton abundance and species composition in the Levantine Sea in November 1991. Stazione Zoologica Anton Dohrn, doi:10.1594/PANGAEA.703972

Mazzocchi, Maria Grazia (2008): Mesozooplankton abundance and species composition in the Sicily Channel in October 1991. Part 2. Stazione Zoologica Anton Dohrn, doi:10.1594/PANGAEA.703256

Ramfos, A. and Isari, S. and Rastaman, N., Mesozooplankton abundance in water of the Ionian Sea (March 2000), Department of Biology, University of Patras, 2008.

Siokou-Frangou, Ioanna et al. (2008): Mesozooplankton abundance in waters of the Aegean Sea at Station O91-GN3619910270126804wp3. Hellenic Center of Marine Research, Institut of Oceanography, Greece, doi:10.1594/PANGAEA.692018

Siokou-Frangou, Ioanna et al. (2008): Mesozooplankton abundance in waters of the Aegean Sea at Station O91-GN3619910270126811wp3. Hellenic Center of Marine Research, Institut of Oceanography, Greece, doi:10.1594/PANGAEA.692019

Siokou-Frangou, Ioanna; Christou, Epaminondas; Giannakourou, Antonia; Zoulias, Theodoros (2008): Mesozooplankton abundance and biomass in surface waters of the Aegean Sea in spring 1997. Station MARCH-1997-GN36199704601MSB01wp2. Hellenic Center of Marine Research, Institut of Oceanography, Greece, doi:10.1594/PANGAEA.688659

Siokou-Frangou, Ioanna; Christou, Epaminondas; Giannakourou, Antonia; Zoulias, Theodoros (2008): Mesozooplankton abundance and biomass in surface waters of the Aegean Sea in spring 1997. Station MARCH-1997-GN36199704601MSB02wp2. Hellenic Center of Marine Research, Institut of Oceanography, Greece, doi:10.1594/PANGAEA.688664

Siokou-Frangou, Ioanna; Christou, Epaminondas; Giannakourou, Antonia; Zoulias, Theodoros (2008): Mesozooplankton abundance and biomass in surface waters of the Aegean Sea in spring 1997. Station MARCH-1997-GN36199704601MSB06wp2. Hellenic Center of Marine Research, Institut of Oceanography, Greece, doi:10.1594/PANGAEA.688665

Siokou-Frangou, Ioanna; Christou, Epaminondas; Giannakourou, Antonia; Zoulias, Theodoros (2008): Mesozooplankton abundance and biomass in surface waters of the Aegean Sea in spring 1997. Station MARCH-1997-GN36199704601MSB07wp2. Hellenic Center of Marine Research, Institut of Oceanography, Greece, doi:10.1594/PANGAEA.688666

Siokou-Frangou, Ioanna; Christou, Epaminondas; Giannakourou, Antonia; Zoulias, Theodoros (2008): Mesozooplankton abundance and biomass in surface waters of the Aegean Sea in spring 1997. Station MARCH-1997-GN36199704603MNB01wp2. Hellenic Center of Marine Research, Institut of Oceanography, Greece, doi:10.1594/PANGAEA.688667

Siokou-Frangou, Ioanna; Christou, Epaminondas; Giannakourou, Antonia; Zoulias, Theodoros (2008): Mesozooplankton abundance and biomass in surface waters of the Aegean Sea in spring 1997. Station MARCH-1997-GN36199704603MNB02wp2. Hellenic Center of Marine Research, Institut of Oceanography, Greece, doi:10.1594/PANGAEA.688734

Siokou-Frangou, Ioanna; Christou, Epaminondas; Giannakourou, Antonia; Zoulias, Theodoros (2008): Mesozooplankton abundance and biomass in surface waters of the Aegean Sea in spring 1997. Station MARCH-1997-GN36199704603MNB03wp2. Hellenic Center of Marine Research, Institut of Oceanography, Greece, doi:10.1594/PANGAEA.688735

Siokou-Frangou, Ioanna; Christou, Epaminondas; Giannakourou, Antonia; Zoulias, Theodoros (2008): Mesozooplankton abundance and biomass in surface waters of the Aegean Sea in spring 1997. Station MARCH-1997-GN36199704603MNB05wp2. Hellenic Center of Marine Research, Institut of Oceanography, Greece, doi:10.1594/PANGAEA.688737

Siokou-Frangou, Ioanna; Christou, Epaminondas; Giannakourou, Antonia; Zoulias, Theodoros (2008): Mesozooplankton abundance and biomass in surface waters of the Aegean Sea in spring 1997. Station MARCH-1997-GN36199704603MNB07wp2. Hellenic Center of Marine Research, Institut of Oceanography, Greece, doi:10.1594/PANGAEA.688739

Siokou-Frangou, Ioanna; Christou, Epaminondas; Rastaman, Nina (2008): Mesozooplankton abundance in waters of the Aegean Sea at Station A92-GN3619920270226404wp2. Hellenic Center of Marine Research, Institut of Oceanography, Greece, doi:10.1594/PANGAEA.693561

Siokou-Frangou, Ioanna; Christou, Epaminondas; Rastaman, Nina (2008): Mesozooplankton abundance in waters of the Aegean Sea at Station A92-GN3619920270226605wp2. Hellenic Center of Marine Research, Institut of Oceanography, Greece, doi:10.1594/PANGAEA.693563

Siokou-Frangou, Ioanna; Christou, Epaminondas; Rastaman, Nina (2008): Mesozooplankton abundance in waters of the Aegean Sea at Station A92-GN3619920270226704wp2. Hellenic Center of Marine Research, Institut of Oceanography, Greece, doi:10.1594/PANGAEA.693564

Siokou-Frangou, Ioanna; Christou, Epaminondas; Rastaman, Nina (2008): Mesozooplankton abundance in waters of the Levantine Sea at Station A92-GN3619920270224502wp2. Hellenic Center of Marine Research, Institut of Oceanography, Greece, doi:10.1594/PANGAEA.693555

Siokou-Frangou, Ioanna; Christou, Epaminondas; Rastaman, Nina (2008): Mesozooplankton abundance in waters of the Levantine Sea at Station A92-GN3619920270224603wp2. Hellenic Center of Marine Research, Institut of Oceanography, Greece, doi:10.1594/PANGAEA.693556

Siokou-Frangou, Ioanna; Christou, Epaminondas; Rastaman, Nina (2008): Mesozooplankton abundance in waters of the Levantine Sea at Station A92-GN3619920270225810wp2. Hellenic Center of Marine Research, Institut of Oceanography, Greece, doi:10.1594/PANGAEA.693559

Siokou-Frangou, Ioanna; Christou, Epaminondas; Rastaman, Nina (2008): Mesozooplankton abundance in waters of the Levantine Sea at Station A92-GN3619920270226804wp2. Hellenic Center of Marine Research, Institut of Oceanography, Greece, doi:10.1594/PANGAEA.693565

Siokou-Frangou, Ioanna; Christou, Epaminondas; Rastaman, Nina (2008): Mesozooplankton abundance in waters of the Levantine Sea at Station A92-GN3619920270226811wp2. Hellenic Center of Marine Research, Institut of Oceanography, Greece, doi:10.1594/PANGAEA.693566

Siokou-Frangou, Ioanna; Christou, Epaminondas; Zervoudaki, Soultana; Zoulias, Theodoros (2008): Mesozooplankton abundance and biomass in waters of the Aegean Sea in September 1997. Station SEPT-1997-GN36199704605MSB01wp2. Hellenic Center of Marine Research, Institut of Oceanography, Greece, doi:10.1594/PANGAEA.690849

Siokou-Frangou, Ioanna; Christou, Epaminondas; Zervoudaki, Soultana; Zoulias, Theodoros (2008): Mesozooplankton abundance and biomass in waters of the Aegean Sea in September 1997. Station SEPT-1997-GN36199704605MSB02wp2. Hellenic Center of Marine Research, Institut of Oceanography, Greece, doi:10.1594/PANGAEA.690850

Siokou-Frangou, Ioanna; Christou, Epaminondas; Zervoudaki, Soultana; Zoulias, Theodoros (2008): Mesozooplankton abundance and biomass in waters of the Aegean Sea in September 1997. Station SEPT-1997-GN36199704605MSB03wp2. Hellenic Center of Marine Research, Institut of Oceanography, Greece, doi:10.1594/PANGAEA.690851

Siokou-Frangou, Ioanna; Christou, Epaminondas; Zervoudaki, Soultana; Zoulias, Theodoros (2008): Mesozooplankton abundance and biomass in waters of the Aegean Sea in September 1997. Station SEPT-1997-GN36199704605MSB06wp2. Hellenic Center of Marine Research, Institut of Oceanography, Greece, doi:10.1594/PANGAEA.690852

Siokou-Frangou, Ioanna; Christou, Epaminondas; Zervoudaki, Soultana; Zoulias, Theodoros (2008): Mesozooplankton abundance and biomass in waters of the Aegean Sea in September 1997. Station SEPT-1997-GN36199704605MSB07wp2. Hellenic Center of Marine Research, Institut of Oceanography, Greece, doi:10.1594/PANGAEA.690853

Siokou-Frangou, Ioanna; Christou, Epaminondas; Zervoudaki, Soultana; Zoulias, Theodoros (2008): Mesozooplankton abundance and biomass in waters of the Aegean Sea in September 1997. Station SEPT-1997-GN36199704606MNB01wp2. Hellenic Center of Marine Research, Institut of Oceanography, Greece, doi:10.1594/PANGAEA.690813

Siokou-Frangou, Ioanna; Christou, Epaminondas; Zervoudaki, Soultana; Zoulias, Theodoros (2008): Mesozooplankton abundance and biomass in waters of the Aegean Sea in September 1997. Station SEPT-1997-GN36199704606MNB02wp2. Hellenic Center of Marine Research, Institut of Oceanography, Greece, doi:10.1594/PANGAEA.690814

Siokou-Frangou, Ioanna; Christou, Epaminondas; Zervoudaki, Soultana; Zoulias, Theodoros (2008): Mesozooplankton abundance and biomass in waters of the Aegean Sea in September 1997. Station SEPT-1997-GN36199704606MNB03wp2. Hellenic Center of Marine Research, Institut of Oceanography, Greece, doi:10.1594/PANGAEA.690815

Siokou-Frangou, Ioanna; Christou, Epaminondas; Zervoudaki, Soultana; Zoulias, Theodoros (2008): Mesozooplankton abundance and biomass in waters of the Aegean Sea in September 1997. Station SEPT-1997-GN36199704606MNB05wp2. Hellenic Center of Marine Research, Institut of Oceanography, Greece, doi:10.1594/PANGAEA.690817

Siokou-Frangou, Ioanna; Christou, Epaminondas; Zervoudaki, Soultana; Zoulias, Theodoros (2008): Mesozooplankton abundance and biomass in waters of the Aegean Sea in September 1997. Station SEPT-1997-GN36199704606MNB07wp2. Hellenic Center of Marine Research, Institut of Oceanography, Greece, doi:10.1594/PANGAEA.690820

Siokou-Frangou, Ioanna; Christou, et al. (2008): Mesozooplankton abundance in waters of the Aegean Sea at Station O91-GN3619910270125307wp3. Hellenic Center of Marine Research, Institut of Oceanography, Greece, doi:10.1594/PANGAEA.691996

Siokou-Frangou, Ioanna; et al. (2008): Mesozooplankton abundance in waters of the Aegean Sea at Station O91-GN3619910270126404wp3. Hellenic Center of Marine Research, Institut of Oceanography, Greece, doi:10.1594/PANGAEA.692014

Siokou-Frangou, Ioanna; et al. (2008): Mesozooplankton abundance in waters of the Aegean Sea at Station O91-GN3619910270126502wp3. Hellenic Center of Marine Research, Institut of Oceanography, Greece, doi:10.1594/PANGAEA.692015

Siokou-Frangou, Ioanna; et al. (2008): Mesozooplankton abundance in waters of the Aegean Sea at Station O91-GN3619910270126605wp3B. Hellenic Center of Marine Research, Institut of Oceanography, Greece, doi:10.1594/PANGAEA.692016

Siokou-Frangou, Ioanna; et al. (2008): Mesozooplankton abundance in waters of the Aegean Sea at Station O91-GN3619910270126704wp3. Hellenic Center of Marine Research, Institut of Oceanography, Greece, doi:10.1594/PANGAEA.692017

Siokou-Frangou, Ioanna; et al. (2008): Mesozooplankton abundance in waters of the Ionian Sea at Station O91-GN3619910270124303wp3. Hellenic Center of Marine Research, Institut of Oceanography, Greece, doi:10.1594/PANGAEA.692870

Siokou-Frangou, Ioanna; et al. (2008): Mesozooplankton abundance in waters of the Ionian Sea at Station O91-GN3619910270124405wp3. Hellenic Center of Marine Research, Institut of Oceanography, Greece, doi:10.1594/PANGAEA.692871

Siokou-Frangou, Ioanna; et al. (2008): Mesozooplankton abundance in waters of the Ionian Sea at Station O91-GN3619910270124502wp3. Hellenic Center of Marine Research, Institut of Oceanography, Greece, doi:10.1594/PANGAEA.692872

Siokou-Frangou, Ioanna; et al. (2008): Mesozooplankton abundance in waters of the Ionian Sea at Station O91-GN3619910270124702wp3. Hellenic Center of Marine Research, Institut of Oceanography, Greece, doi:10.1594/PANGAEA.692874

Siokou-Frangou, Ioanna; et al. (2008): Mesozooplankton abundance in waters of the Ionian Sea at Station O91-GN3619910270124830wp3. Hellenic Center of Marine Research, Institut of Oceanography, Greece, doi:10.1594/PANGAEA.692875

Siokou-Frangou, Ioanna; et al. (2008): Mesozooplankton abundance in waters of the Ionian Sea at Station O91-GN3619910270125202wp3. Hellenic Center of Marine Research, Institut of Oceanography, Greece, doi:10.1594/PANGAEA.692876

Siokou-Frangou, Ioanna; et al. (2008): Mesozooplankton abundance in waters of the Ionian Sea at Station O91-GN3619910270125810wp3. Hellenic Center of Marine Research, Institut of Oceanography, Greece, doi:10.1594/PANGAEA.692877

Siokou-Frangou, Ioanna; et al. (2008): Mesozooplankton abundance in waters of the Ionian Sea at Station O91-GN3619910270125817wp3. Hellenic Center of Marine Research, Institut of Oceanography, Greece, doi:10.1594/PANGAEA.692878

Siokou-Frangou, Ioanna; et al. (2008): Mesozooplankton abundance in waters of the Ionian Sea at Station O91-GN3619910270126002wp3. Hellenic Center of Marine Research, Institut of Oceanography, Greece, doi:10.1594/PANGAEA.692240

Siokou-Frangou, Ioanna; et al. (2008): Mesozooplankton abundance in waters of the Ionian Sea at Station O91-GN3619910270126102wp3B, doi:10.1594/PANGAEA.692241

Siokou-Frangou, Ioanna; et al. (2008): Mesozooplankton abundance in waters of the Ionian Sea at Station O91-GN3619910270126203wp3., doi:10.1594/PANGAEA.692868

Siokou-Frangou, Ioanna; et al. (2008): Mesozooplankton abundance in waters of the Ionian Sea at Station O91-GN3619910270136903wp3. Hellenic Center of Marine Research, Institut of Oceanography, Greece, doi:10.1594/PANGAEA.692869

Siokou-Frangou, Ioanna; et al.(2008): Mesozooplankton abundance in waters of the Ionian Sea at Station O91-GN3619910270124603wp3. Hellenic Center of Marine Research, Institut of Oceanography, Greece, doi:10.1594/PANGAEA.692873

Siokou-Frangou, Ioanna; Zervoudaki, Soultana; Christou, Epaminondas; Zoulias, Theodoros (2008): Mesozooplankton abundance in water of the Aegean Sea at Station MAY-1997-MNB2wp2. Hellenic Center of Marine Research, Institut of Oceanography, Greece, doi:10.1594/PANGAEA.695140

Siokou-Frangou, Ioanna; Zervoudaki, Soultana; Christou, Epaminondas; Zoulias, Theodoros (2008): Mesozooplankton abundance in water of the Aegean Sea at Station MAY-1997-MNB5wp2. Hellenic Center of Marine Research, Institut of Oceanography, Greece, doi:10.1594/PANGAEA.695144

Siokou-Frangou, Ioanna; Zervoudaki, Soultana; Christou, Epaminondas; Zoulias, Theodoros (2008): Mesozooplankton abundance in water of the Aegean Sea at Station MAY-1997-MNB7wp2. Hellenic Center of Marine Research, Institut of Oceanography, Greece, doi:10.1594/PANGAEA.695146

Zervoudaki, Soultana; Christou, Epaminondas; Siokou-Frangou, Ioanna; Zoulias, Theodoros (2008): Mesozooplankton abundance in water of the Aegean Sea at Station SEPT-1998-MNB5wp2. Hellenic Center of Marine Research, Institut of Oceanography, Greece, doi:10.1594/PANGAEA.695158

Zervoudaki, Soultana; Christou, Epaminondas; Siokou-Frangou, Ioanna; Zoulias, Theodoros (2008): Mesozooplankton abundance in water of the Ionian Sea at Station SEPT-2000-IKO3wp2. Hellenic Center of Marine Research, Institut of Oceanography, Greece, doi:10.1594/PANGAEA.695161

Zervoudaki, Soultana; Christou, Epaminondas; Siokou-Frangou, Ioanna; Zoulias, Theodoros (2008): Mesozooplankton abundance in water of the Ionian Sea at Station SEPT-2000-IRI47wp2. Hellenic Center of Marine Research, Institut of Oceanography, Greece, doi:10.1594/PANGAEA.695186

Zervoudaki, Soultana; Siokou-Frangou, Ioanna; Christou, Epaminondas; Zoulias, Theodoros (2008): Mesozooplankton abundance in water of the Aegean Sea at Station

JUNE-1998-MNB2wp2. Hellenic Center of Marine Research, Institut of Oceanography, Greece, doi:10.1594/PANGAEA.695102

Zervoudaki, Soultana; Siokou-Frangou, Ioanna; Christou, Epaminondas; Zoulias, Theodoros (2008): Mesozooplankton abundance in water of the Aegean Sea in June 1998. Hellenic Center of Marine Research, Institut of Oceanography, Greece, doi:10.1594/PANGAEA.695104

Koppelmann, Rolf; Weikert, Horst (2008): Plankton abundance of mocness net M44/4_D-MOC216. doi:10.1594/PANGAEA.249974

Koppelmann, Rolf; Weikert, Horst (2008): Plankton abundance of mocness net M44/4_D-MOC220. doi:10.1594/PANGAEA.249975

Koppelmann, Rolf; Weikert, Horst (2008): Plankton abundance of mocness net M44/4_D-MOC242. doi:10.1594/PANGAEA.249977

Koppelmann, Rolf; Weikert, Horst (2008): Plankton abundance of mocness net M44/4_D-MOC249. doi:10.1594/PANGAEA.249978

Koppelmann, Rolf; Weikert, Horst (2008): Plankton abundance of mocness net M44/4_D-MOC268. doi:10.1594/PANGAEA.81995

Koppelmann, Rolf; Weikert, Horst (2008): Plankton abundance of mocness net M44/4_D-MOC273. doi:10.1594/PANGAEA.81998

Koppelmann, Rolf; Weikert, Horst (2008): Plankton abundance of mocness net M44/4_D-MOC280. doi:10.1594/PANGAEA.81999

Koppelmann, Rolf; Weikert, Horst (2008): Plankton abundance of mocness net M44/4_D-MOC281. doi:10.1594/PANGAEA.249979

Koppelmann, Rolf; Weikert, Horst (2008): Plankton abundance of mocness net M44/4_D-MOC293. doi:10.1594/PANGAEA.249980

Koppelmann, Rolf; Weikert, Horst (2008): Plankton abundance of mocness net M44/4_D-MOC304. doi:10.1594/PANGAEA.249982

Mazzocchi, Maria Grazia (2008): Mesozooplankton abundance and species composition in the Ionian Sea in April-May 1992. Part 2. Stazione Zoologica Anton Dohrn, doi:10.1594/PANGAEA.703258

Mazzocchi, Maria Grazia (2008): Mesozooplankton abundance and species composition in the Ionian Sea in April-May 1999. Stazione Zoologica Anton Dohrn, doi:10.1594/PANGAEA.703201

Mazzocchi, Maria Grazia (2008): Mesozooplankton abundance and species composition in the Levantine Sea in November 1991. Stazione Zoologica Anton Dohrn, doi:10.1594/PANGAEA.703972

Mazzocchi, Maria Grazia (2008): Mesozooplankton abundance and species composition in the Sicily Channel in October 1991. Part 2. Stazione Zoologica Anton Dohrn, doi:10.1594/PANGAEA.703256

Ramfos, A. and Isari, S. and Rastaman, N., Mesozooplankton abundance in water of the Ionian Sea (March 2000), Department of Biology, University of Patras, 2008.

Siokou-Frangou, Ioanna et al. (2008): Mesozooplankton abundance in waters of the Aegean Sea at Station O91-GN3619910270126804wp3. Hellenic Center of Marine Research, Institut of Oceanography, Greece, doi:10.1594/PANGAEA.692018

Siokou-Frangou, Ioanna et al. (2008): Mesozooplankton abundance in waters of the Aegean Sea at Station O91-GN3619910270126811wp3. Hellenic Center of Marine Research, Institut of Oceanography, Greece, doi:10.1594/PANGAEA.692019

Siokou-Frangou, Ioanna; Christou, Epaminondas; Rastaman, Nina (2008): Mesozooplankton abundance in waters of the Aegean Sea at Station A92-GN3619920270226404wp2. Hellenic Center of Marine Research, Institut of Oceanography, Greece, doi:10.1594/PANGAEA.693561

Siokou-Frangou, Ioanna; Christou, Epaminondas; Rastaman, Nina (2008): Mesozooplankton abundance in waters of the Aegean Sea at Station A92-GN3619920270226605wp2. Hellenic Center of Marine Research, Institut of Oceanography, Greece, doi:10.1594/PANGAEA.693563

Siokou-Frangou, Ioanna; Christou, Epaminondas; Rastaman, Nina (2008): Mesozooplankton abundance in waters of the Aegean Sea at Station A92-GN3619920270226704wp2. Hellenic Center of Marine Research, Institut of Oceanography, Greece, doi:10.1594/PANGAEA.693564

Siokou-Frangou, Ioanna; Christou, Epaminondas; Rastaman, Nina (2008): Mesozooplankton abundance in waters of the Levantine Sea at Station A92-GN3619920270224502wp2. Hellenic Center of Marine Research, Institut of Oceanography, Greece, doi:10.1594/PANGAEA.693555

Siokou-Frangou, Ioanna; Christou, Epaminondas; Rastaman, Nina (2008): Mesozooplankton abundance in waters of the Levantine Sea at Station A92-GN3619920270224603wp2. Hellenic Center of Marine Research, Institut of Oceanography, Greece, doi:10.1594/PANGAEA.693556

Siokou-Frangou, Ioanna; Christou, Epaminondas; Rastaman, Nina (2008): Mesozooplankton abundance in waters of the Levantine Sea at Station A92-GN3619920270225810wp2. Hellenic Center of Marine Research, Institut of Oceanography, Greece, doi:10.1594/PANGAEA.693559

Siokou-Frangou, Ioanna; Christou, Epaminondas; Rastaman, Nina (2008): Mesozooplankton abundance in waters of the Levantine Sea at Station A92-

GN3619920270226804wp2. Hellenic Center of Marine Research, Institut of Oceanography, Greece, doi:10.1594/PANGAEA.693565

Siokou-Frangou, Ioanna; Christou, Epaminondas; Rastaman, Nina (2008): Mesozooplankton abundance in waters of the Levantine Sea at Station A92-GN3619920270226811wp2. Hellenic Center of Marine Research, Institut of Oceanography, Greece, doi:10.1594/PANGAEA.693566

Siokou-Frangou, Ioanna; Christou, Epaminondas; Zervoudaki, Soultana; Zoulias, Theodoros (2008): Mesozooplankton abundance and biomass in waters of the Aegean Sea in September 1997. Station SEPT-1997-GN36199704605MSB01wp2. Hellenic Center of Marine Research, Institut of Oceanography, Greece, doi:10.1594/PANGAEA.690849

Siokou-Frangou, Ioanna; Christou, Epaminondas; Zervoudaki, Soultana; Zoulias, Theodoros (2008): Mesozooplankton abundance and biomass in waters of the Aegean Sea in September 1997. Station SEPT-1997-GN36199704605MSB02wp2. Hellenic Center of Marine Research, Institut of Oceanography, Greece, doi:10.1594/PANGAEA.690850

Siokou-Frangou, Ioanna; Christou, Epaminondas; Zervoudaki, Soultana; Zoulias, Theodoros (2008): Mesozooplankton abundance and biomass in waters of the Aegean Sea in September 1997. Station SEPT-1997-GN36199704605MSB03wp2. Hellenic Center of Marine Research, Institut of Oceanography, Greece, doi:10.1594/PANGAEA.690851

Siokou-Frangou, Ioanna; Christou, Epaminondas; Zervoudaki, Soultana; Zoulias, Theodoros (2008): Mesozooplankton abundance and biomass in waters of the Aegean Sea in September 1997. Station SEPT-1997-GN36199704605MSB06wp2. Hellenic Center of Marine Research, Institut of Oceanography, Greece, doi:10.1594/PANGAEA.690852

Siokou-Frangou, Ioanna; Christou, Epaminondas; Zervoudaki, Soultana; Zoulias, Theodoros (2008): Mesozooplankton abundance and biomass in waters of the Aegean Sea in September 1997. Station SEPT-1997-GN36199704605MSB07wp2. Hellenic Center of Marine Research, Institut of Oceanography, Greece, doi:10.1594/PANGAEA.690853

Siokou-Frangou, Ioanna; Christou, Epaminondas; Zervoudaki, Soultana; Zoulias, Theodoros (2008): Mesozooplankton abundance and biomass in waters of the Aegean Sea in September 1997. Station SEPT-1997-GN36199704606MNB01wp2. Hellenic Center of Marine Research, Institut of Oceanography, Greece, doi:10.1594/PANGAEA.690813

Siokou-Frangou, Ioanna; Christou, Epaminondas; Zervoudaki, Soultana; Zoulias, Theodoros (2008): Mesozooplankton abundance and biomass in waters of the Aegean Sea in September 1997. Station SEPT-1997-GN36199704606MNB02wp2. Hellenic Center of Marine Research, Institut of Oceanography, Greece, doi:10.1594/PANGAEA.690814

Siokou-Frangou, Ioanna; Christou, Epaminondas; Zervoudaki, Soultana; Zoulias, Theodoros (2008): Mesozooplankton abundance and biomass in waters of the Aegean Sea in September 1997. Station SEPT-1997-GN36199704606MNB03wp2. Hellenic Center of Marine Research, Institut of Oceanography, Greece, doi:10.1594/PANGAEA.690815

Siokou-Frangou, Ioanna; Christou, Epaminondas; Zervoudaki, Soultana; Zoulias, Theodoros (2008): Mesozooplankton abundance and biomass in waters of the Aegean Sea in September 1997. Station SEPT-1997-GN36199704606MNB05wp2. Hellenic Center of Marine Research, Institut of Oceanography, Greece, doi:10.1594/PANGAEA.690817

Siokou-Frangou, Ioanna; Christou, Epaminondas; Zervoudaki, Soultana; Zoulias, Theodoros (2008): Mesozooplankton abundance and biomass in waters of the Aegean Sea in September 1997. Station SEPT-1997-GN36199704606MNB07wp2. Hellenic Center of Marine Research, Institut of Oceanography, Greece, doi:10.1594/PANGAEA.690820

Siokou-Frangou, Ioanna; Christou, et al. (2008): Mesozooplankton abundance in waters of the Aegean Sea at Station O91-GN3619910270125307wp3. Hellenic Center of Marine Research, Institut of Oceanography, Greece, doi:10.1594/PANGAEA.691996

Siokou-Frangou, Ioanna; et al. (2008): Mesozooplankton abundance in waters of the Aegean Sea at Station O91-GN3619910270126404wp3. Hellenic Center of Marine Research, Institut of Oceanography, Greece, doi:10.1594/PANGAEA.692014

Siokou-Frangou, Ioanna; et al. (2008): Mesozooplankton abundance in waters of the Aegean Sea at Station O91-GN3619910270126502wp3. Hellenic Center of Marine Research, Institut of Oceanography, Greece, doi:10.1594/PANGAEA.692015

Siokou-Frangou, Ioanna; et al. (2008): Mesozooplankton abundance in waters of the Aegean Sea at Station O91-GN3619910270126605wp3B. Hellenic Center of Marine Research, Institut of Oceanography, Greece, doi:10.1594/PANGAEA.692016

Siokou-Frangou, Ioanna; et al. (2008): Mesozooplankton abundance in waters of the Aegean Sea at Station O91-GN3619910270126704wp3. Hellenic Center of Marine Research, Institut of Oceanography, Greece, doi:10.1594/PANGAEA.692017

Siokou-Frangou, Ioanna; et al. (2008): Mesozooplankton abundance in waters of the Ionian Sea at Station O91-GN3619910270124303wp3. Hellenic Center of Marine Research, Institut of Oceanography, Greece, doi:10.1594/PANGAEA.692870

Siokou-Frangou, Ioanna; et al. (2008): Mesozooplankton abundance in waters of the Ionian Sea at Station O91-GN3619910270124405wp3. Hellenic Center of Marine Research, Institut of Oceanography, Greece, doi:10.1594/PANGAEA.692871

Siokou-Frangou, Ioanna; et al. (2008): Mesozooplankton abundance in waters of the Ionian Sea at Station O91-GN3619910270124502wp3. Hellenic Center of Marine Research, Institut of Oceanography, Greece, doi:10.1594/PANGAEA.692872

Siokou-Frangou, Ioanna; et al. (2008): Mesozooplankton abundance in waters of the Ionian Sea at Station O91-GN3619910270124702wp3. Hellenic Center of Marine Research, Institut of Oceanography, Greece, doi:10.1594/PANGAEA.692874

Siokou-Frangou, Ioanna; et al. (2008): Mesozooplankton abundance in waters of the Ionian Sea at Station O91-GN3619910270124830wp3. Hellenic Center of Marine Research, Institut of Oceanography, Greece, doi:10.1594/PANGAEA.692875

Siokou-Frangou, Ioanna; et al. (2008): Mesozooplankton abundance in waters of the Ionian Sea at Station O91-GN3619910270125202wp3. Hellenic Center of Marine Research, Institut of Oceanography, Greece, doi:10.1594/PANGAEA.692876

Siokou-Frangou, Ioanna; et al. (2008): Mesozooplankton abundance in waters of the Ionian Sea at Station O91-GN3619910270125810wp3. Hellenic Center of Marine Research, Institut of Oceanography, Greece, doi:10.1594/PANGAEA.692877

Siokou-Frangou, Ioanna; et al. (2008): Mesozooplankton abundance in waters of the Ionian Sea at Station O91-GN3619910270125817wp3. Hellenic Center of Marine Research, Institut of Oceanography, Greece, doi:10.1594/PANGAEA.692878

Siokou-Frangou, Ioanna; et al. (2008): Mesozooplankton abundance in waters of the Ionian Sea at Station O91-GN3619910270126002wp3. Hellenic Center of Marine Research, Institut of Oceanography, Greece, doi:10.1594/PANGAEA.692240

Siokou-Frangou, Ioanna; et al. (2008): Mesozooplankton abundance in waters of the Ionian Sea at Station O91-GN3619910270126102wp3B, doi:10.1594/PANGAEA.692241

Siokou-Frangou, Ioanna; et al. (2008): Mesozooplankton abundance in waters of the Ionian Sea at Station O91-GN3619910270126203wp3., doi:10.1594/PANGAEA.692868

Siokou-Frangou, Ioanna; et al. (2008): Mesozooplankton abundance in waters of the Ionian Sea at Station O91-GN3619910270136903wp3. Hellenic Center of Marine Research, Institut of Oceanography, Greece, doi:10.1594/PANGAEA.692869

Siokou-Frangou, Ioanna; et al.(2008): Mesozooplankton abundance in waters of the Ionian Sea at Station O91-GN3619910270124603wp3. Hellenic Center of Marine Research, Institut of Oceanography, Greece, doi:10.1594/PANGAEA.692873

Siokou-Frangou, Ioanna; Zervoudaki, Soultana; Christou, Epaminondas; Zoulias, Theodoros (2008): Mesozooplankton abundance in water of the Aegean Sea at Station MAY-1997-MNB2wp2. Hellenic Center of Marine Research, Institut of Oceanography, Greece, doi:10.1594/PANGAEA.695140

Siokou-Frangou, Ioanna; Zervoudaki, Soultana; Christou, Epaminondas; Zoulias, Theodoros (2008): Mesozooplankton abundance in water of the Aegean Sea at Station MAY-1997-MNB7wp2. Hellenic Center of Marine Research, Institut of Oceanography, Greece, doi:10.1594/PANGAEA.695146

Zervoudaki, Soultana; Christou, Epaminondas; Siokou-Frangou, Ioanna; Zoulias, Theodoros (2008): Mesozooplankton abundance in water of the Aegean Sea at Station SEPT-1998-MNB5wp2. Hellenic Center of Marine Research, Institut of Oceanography, Greece, doi:10.1594/PANGAEA.695158

Zervoudaki, Soultana; Christou, Epaminondas; Siokou-Frangou, Ioanna; Zoulias, Theodoros (2008): Mesozooplankton abundance in water of the Ionian Sea at Station SEPT-2000-IKO3wp2. Hellenic Center of Marine Research, Institut of Oceanography, Greece, doi:10.1594/PANGAEA.695161

Zervoudaki, Soultana; Christou, Epaminondas; Siokou-Frangou, Ioanna; Zoulias, Theodoros (2008): Mesozooplankton abundance in water of the Ionian Sea at Station SEPT-2000-IRI47wp2. Hellenic Center of Marine Research, Institut of Oceanography, Greece, doi:10.1594/PANGAEA.695186

---

## Author Comment (AC3) · 12 Jun 2020

Thank you for your constructive remarks. Please find our detailed responses to your comments, including expected modifications of the manuscript, below.

1. **COMMENT:** This study presents the distribution of omega saturation state as if this has not been published before. It is not clear if this is novel result of this study or the profiles were constructed based on data published before. If this is not clear, it is difficult to judge the suitability of carbonate chemistry data.

**REPLY:** The  $\Omega_{ar}$  presented in this study is from each station at the time of the collection. This has been made more explicit in the Methods section. In the discussion, we use the aragonite saturation state data in the Mediterranean Sea, and elaborate on the gradient of West -East  $\Omega_{ar}$ . Detailed Mediterranean carbonate chemistry data used in our study from the same oceanographic cruise and stations have being already published (e.g. D'Amario et al., 2017a, 2018; Gemayel et al., 2015; Hassoun et al., 2015a, 2015b; Mallo et al., 2017) and archived in an open access database (https://doi.pangaea.de/10.1594/PANGAEA.841933; Goyet et al., 2015).

Updated text:

The  $\Omega_{ar}$  presented in this study is from data collected during the cruise at the time of plankton sampling.

2. **COMMENT:** By splitting the basin into W-E before analysing the station variability within sub-basin ignores inter- and intra-specific variability of each sub-basin. Based on the abundance data (Fig 5), there intra sub basin variability could be as large as the W-E comparison. The same pattern is true for species distribution. As such, the authors first need to reconcile the level of variability between the stations before they can attempt to make a W-E division.

**3. COMMENT:** Based on Fig 2 and Fig 5, W and E part of the basins have comparable abundances and species distribution, the difference is really in the transition zone between the E and W (station 9, 14, 15, 16). This is the real results

and not random W-E division. However, this makes a large portion of the discussion invalid and needs to be reconsidered and restructured.

**REPLY:** The Mediterranean Sea is separated into two large sub-basins by the relatively shallow (average depth of 330 m) Strait of Sicily. These basins have distinct environmental profiles, with the eastern basin characterised by warmer, more saline and oligotrophic conditions and the western basin characterised by cooler, less saline conditions. The division between the eastern and western Mediterranean Sea sub basins is recognised as an important biogeographical boundary (Dayan et al., 2015; Hassoun et al., 2015b; Rohling et al., 2009; Schneider et al., 2007; Uitz et al., 2012).

In our study, average abundance across all stations in the Mediterranean Sea was 1.17 ind.  $m^{-3} \pm 1.52$  (SD), yet there were clear differences in abundance between the two sub-basins, with an average of 0.4 ind.  $m^{-3} \pm 0.37$  (SD) for the western basin, and an average 5x higher of 1.96 ind.  $m^{-3} \pm 1.8$  (SD) in the eastern basin. Figures 2 and 5 show consistently lower pteropod standing stocks in the Western Mediterranean than in the eastern Mediterranean. In the eastern basin, the overall abundance is higher, as is the variability in abundance. Figure 2 (Figure 1 here) of the manuscript has been re-done to better illustrate the difference in total abundance between the eastern and western basins.

**Figure 1.** Absolute abundance of planktic pteropods from stations 1-22 on the MedSEA cruise, 2013. The category of 'Others' for each family includes specimens that were not a target species in this study or that were unidentifiable to the species level. Total pteropod abundance is greater and more variable in the eastern Mediterranean basin than in the western.

In order to analyse significant differences in total and individual species abundance considering both sub-basins, a Kruskal-Wallis Test was used between western and eastern stations, however this analysis has now been removed and we are use a more station-level approach with a parsimonious CCA and Pearson's correlations. However, due to the substantial difference in environmental parameters between the eastern and western basin, as well as the difference in total pteropod abundance, we still highlight the differences in the Mediterranean in terms of the two major sub-basins. A k-means cluster analysis was initially conducted on the species abundances, (IBM SPSS v23), incrementally increasing the amount of clusters until all variables significantly contributed to the cluster formation (3 clusters). The results show a uniform distribution with a cluster in the western Mediterranean, and more diverse communities in the eastern Mediterranean (Figure 2).